# ON INVARIANCE PENALTIES FOR RISK MINIMIZATION

## ABSTRACT

The Invariant Risk Minimization (IRM) principle was first proposed by Arjovsky et al. (2019) to address the domain generalization problem by leveraging data heterogeneity from differing experimental conditions. Specifically, IRM seeks to find a data representation under which an optimal classifier remains invariant across all domains. Despite the conceptual appeal of IRM, the effectiveness of the originally proposed invariance penalty has recently been brought into question through stylized experiments and counterexamples. In this work, we investigate the relationship between the data representation, invariance penalty, and risk. In doing so, we propose a novel invariance penalty, and utilize it to design an adaptive rule for tuning the coefficient of the penalty proposed by Arjovsky et al. (2019). Moreover, we provide practical insights on how to avoid the potential failure of IRM considered in the nascent counterexamples. Finally, we conduct numerical experiments on both synthetic and real-world data sets with the objective of building invariant predictors. In our non-synthetic experiments, we sought to build a predictor of human health status using a collection of data sets from various studies which investigate the relationship between human gut microbiome and a particular disease. We substantiate the effectiveness of our proposed approach on these data sets and thus further facilitate the adoption of the IRM principle in other real-world applications.

## 1 INTRODUCTION

Under the learning paradigm of Empirical Risk Minimization (ERM) (Vapnik, 1992), data is assumed to consist of independent and identically distributed (iid) samples from an underlying generating distribution. As the data generating distribution is often unknown in practice, ERM seeks predictors with minimal average training error (i.e., empirical risk) over the training set. However, shuffling and treating data as iid risks possibly losing important information about the underlying conditions of the data generating process. Despite becoming a ubiquitous paradigm in machine learning, a growing body of literature (Arjovsky et al., 2019; Teney et al., 2020) has revealed that ERM and the the common practice of shuffling data inadvertently results in capturing all correlations found in the training data, whether spurious or causal, and produces models that fail to *generalize* to test data. The potential variation of experimental conditions that can exist at training time and during deployment in real-world applications, manifests in discrepancies between training and testing distributions. This, in turn, highlights the need for machine learning algorithms to *generalize out-of-distribution (OoD)*.

Shuffling and treating data as iid risks possibly losing important information about the underlying conditions of the data generating process. As will be demonstrated in this work, partitioning training data into *environments*, e.g., based on the conditions under which data is generated, can exploit these differences to enhance generalization. One promising approach based on this observation is that of Arjovsky et al. (2019), in which the principle of Invariant Risk Minimization (IRM) is introduced. The objective of IRM is to find a predictor that is invariant across all training environments (see Definition 1 and Equation 1). Because of the conceptually appealing nature of IRM and its potential to address the OoD-generalization problem, there is a stream of literature scrutinizing various facets of the original framework, e.g., extensions to other settings including online learning (Javed et al., 2020) and treatment effect estimation (Shi et al., 2020), fairness (Adragna et al., 2020), introducing game-theoretic interpretations (Ahuja et al., 2020), and raising concerns on the drawbacks and limitations of current IRM implementations (Rosenfeld et al., 2021; Kamath et al., 2021). For an in-depth overview of the broader generalization literature, we refer the interested reader to (Arjovsky, 2020) and the references therein, and for an empirical evaluation of the performance of a number of the state-of-the-art methods on various test cases, we refer the reader to (Gulrajani & Lopez-Paz, 2020).

In this paper, we introduce two practical implementations of the IRM principal to increase its appeal and applicability in real-world settings. First, we propose an invariance penalty that is directly related to risk. More precisely, we show that the risk in each environment under an *arbitrary* classifier equals to the risk under the *optimal* classifier for that environment plus the newly proposed invariance penalty between the said classifier and the optimal one. Second, we build on these initial findings to provide practitioners who currently use the original IRMv1 implementation with an adaptive rule by which to choose the penalty coefficient appropriately. In doing so, we characterize the difference between our proposed invariance penalty and the one proposed by Arjovsky et al. (2019) in terms of the eigenvalues of the Gram matrix of the data representation. This eigenstructure plays a significant role in the failure of invariance penalties including the one proposed by Arjovsky et al. (2019).

In addition to providing practitioners with two valid approaches for implementing IRM, this work serves to illustrate the importance of the eigenstructure of the Gram matrix of the data representation for IRM. In particular, we revisit the counterexample of Rosenfeld et al. (2021) where the invariance penalty of Arjovsky et al. (2019) can be made arbitrarily small for a non-invariant representation. We show that the Gram matrix is ill-conditioned in such cases. We then provide a practical solution to alleviate such behavior, in particular for the case where the representation is parameterized by a neural network. Moreover, we show that the proposed framework finds an invariant predictor for the setting in which the data is generated according to a linear Structural Equation Model (SEM) when provided a sufficient number of training environments under a mild non-degeneracy condition, which is similar in nature to the ones considered in (Arjovsky et al., 2019; Rosenfeld et al., 2021). Finally, we evaluate our method on various test cases including InvarianceUnitTests (Aubin et al., 2020) and HealthyGutTests. InvarianceUnitTests is a test bed with three synthetically generated data sets capturing different structures of spurious correlations. HealthyGutTests is a curated collection of biomedical data sets based on a prior meta-analysis of various microbiome studies (Gupta et al., 2020) in which the relationships between human gut microbiome composition and various disease phenotypes can be investigated.

The remainder of the paper is organized as follows. In Section 2, we formally define the notion of invariant prediction, the invariant risk minimization principle, and its relaxation proposed by Arjovsky et al. (2019). In Sections 3 and 4, we introduce our more practical implementation and the rationale for its design. In Section 5, we evaluate the efficacy of our proposed model and compare it with other variations of IRM over a series of experiments. We conclude the paper in Section 6. All mathematical proofs are presented in the Appendix.

## 2 BACKGROUND: INVARIANT PREDICTION

In this paper, we consider data $(X^e, Y^e)$ collected from multiple training environments $e \in \mathcal{E}_{\mathrm{tr}}$ where the distribution of $(X^i, Y^i)$ and $(X^j, Y^j)$ may be different for $i \neq j$ with $i, j \in \mathcal{E}_{\mathrm{tr}}$. We denote by $R_e$ the risk under environment $e$. That is, for predictor $f : \mathcal{X} \to \mathcal{Y}$, and loss function $\ell : \mathcal{Y} \times \mathcal{Y} \to \mathbb{R}$, the risk under environment $e$ is defined as $R_e(f) = \mathbf{E}_{X^e, Y^e}[\ell(f(X^e), Y^e)]$.

### 2.1 INVARIANT RISK MINIMIZATION

Arjovsky et al. (2019) define the notion of invariant predictors under a multi-environment setting as follows.

**Definition 1** (Invariant Predictor). A data representation $\varphi : \mathcal{X} \to \mathcal{H}$ is said to elicit an invariant predictor $w \circ \varphi$ across environments $\mathcal{E}$ if there exists a classifier $w : \mathcal{H} \to \mathcal{Y}$, which is optimal for all environments, i.e., $w \in \mathrm{argmin}_{\tilde{w}:\mathcal{H}\to\mathcal{Y}} R_e(\tilde{w} \circ \varphi)$ for all $e \in \mathcal{E}$.

To find such invariant predictors, Arjovsky et al. (2019) introduce the notion of the Invariant Risk Minimization (IRM) principle:

$$
\min_{\substack{\varphi:\mathcal{X}\to\mathcal{H} \\ w:\mathcal{H}\to\mathcal{Y}}} \quad \sum_{e\in\mathcal{E}_{\mathrm{tr}}} R_e(w \circ \varphi)
$$

$$
\text{subject to} \quad w \in \mathrm{argmin}_{\tilde{w}:\mathcal{H}\to\mathcal{Y}} R_e(\tilde{w} \circ \varphi), \ \forall e \in \mathcal{E}_{\mathsf{tr}}.
$$

(1)

As this bi-leveled optimization problem is rather intractable, Arjovsky et al. (2019) propose a practical implementation of IRM by relaxing the invariance constraint (which itself requires solving an optimization problem) to an invariance penalty. We review its derivation in what follows.

## 2.2 IRMv1: A Relaxation of IRM

In order to provide an implementation of IRM, Arjovsky et al. (2019) restrict the classifier $w$ to linear functions, i.e.,

$$\min_{\substack{\varphi:\mathcal{X}\to\mathcal{H} \\ w\in\mathbb{R}^{d_\varphi}}} \quad \sum_{e\in\mathcal{E}_{\mathrm{tr}}} R_e\left(w^\top\varphi\right)$$

$$\text{subject to} \quad w\in\operatorname*{argmin}_{\tilde{w}\in\mathbb{R}^{d_\varphi}} R_e\left(\tilde{w}^\top\varphi\right),\ \forall e\in\mathcal{E}_{\mathrm{tr}}. \tag{2}$$

To motivate their proposed penalty, Arjovsky et al. (2019) first consider the squared loss, i.e., $\ell(f(x),y)=\|f(x)-y\|^2$ where $\|\cdot\|$ denotes the Euclidean norm. Define matrix $\mathcal{I}_e(\varphi)$ as

$$\mathcal{I}_e(\varphi) := \mathbf{E}_{X^e}\left[\varphi(X^e)\varphi(X^e)^\top\right]. \tag{3}$$

Assuming that $\mathcal{I}_e(\varphi)$ is full rank for a fixed $\varphi$, its respective optimal classifier is unique, i.e., $\operatorname{argmin}_{\tilde{w}\in\mathbb{R}^{d_\varphi}} R_e\left(\tilde{w}^\top\varphi\right)=w_e^\star(\varphi)$ where

$$w_e^\star(\varphi) := \mathcal{I}_e(\varphi)^{-1}\mathbf{E}_{X^e,Y^e}\left[\varphi(X^e)Y^e\right]. \tag{4}$$

Hence, in this setting, the invariant constraint of IRM in equation 2 can be simplified to $w=w_e^\star(\varphi)$ for all $e\in\mathcal{E}_{\mathrm{tr}}$. A natural relaxation of the constraint $w-w_e^\star(\varphi)=0$ to a penalty is $\|w-w_e^\star(\varphi)\|^2$.[1] However, Arjovsky et al. (2019) show that this penalty may not capture invariance by constructing an example for which $\|w-w_e^\star(\varphi)\|^2$ is not well-behaved (see Section 4.3 for more details). They argue that undoing the matrix inversion $\mathcal{I}_e(\varphi)^{-1}$, which appears in the computation of $w_e^\star(\varphi)$ could improve the behavior of the invariance penalty. That is, considering $\|\mathcal{I}_e(\varphi)(w-w_e^\star(\varphi))\|^2$ as the invariance penalty. Moreover, for the squared loss, one can show that

$$\|\mathcal{I}_e(\varphi)(w-w_e^\star(\varphi))\|^2 = (1/4)\left\|\nabla_w R_e(w^\top\varphi)\right\|^2. \tag{5}$$

Hence, Arjovsky et al. (2019) propose the following invariance penalty

$$\rho_e^{\mathrm{IRMv1}}(\varphi,w) := \left\|\nabla_w R_e(w^\top\varphi)\right\|^2. \tag{6}$$

Using the penalty equation 6, the relaxation of IRM is given by

$$\min_{\varphi,\,w}\ \sum_{e\in\mathcal{E}_{\mathrm{tr}}} R_e(w^\top\varphi) + \lambda\rho_e^{\mathrm{IRMv1}}(\varphi,w), \tag{7}$$

where $\lambda\geq 0$ is the penalty coefficient. Notice that for a given $w$ and $\varphi$, the predictor $w\circ\varphi$ can be expressed using different classifiers and data representations, i.e., $w\circ\varphi=\tilde{w}\circ\tilde{\varphi}$ where $\tilde{w}=w\circ\psi^{-1}$ and $\tilde{\varphi}=\psi\circ\varphi$ for some invertible mapping $\psi:\mathcal{H}\to\mathcal{H}$. Hence, in principle, it is possible to fix $w$ without loss of generality. By relying on this observation, Arjovsky et al. (2019) fix the classifier as a scalar $w=1$, and, thus, search for an invariant data representation of the form $\varphi\in\mathbb{R}^{1\times d_x}$. Their relaxation of IRM, which they refer to by IRMv1 is given by

$$\min_{\varphi}\ \sum_{e\in\mathcal{E}_{\mathrm{tr}}} R_e\left(\varphi\right) + \lambda\rho_e^{\mathrm{IRMv1}}(\varphi,1.0). \tag{IRMv1}$$

Although equation 5 only holds for squared loss, Arjovsky et al. (2019, Theorem 4) justify the choice of $\|\nabla_{w|w=1.0}R_e(w^\top\varphi)\|^2$ as an invariance penalty for other loss functions, e.g., cross-entropy loss. More precisely, let $\Phi$ be the matrix that parameterizes the data representation. They show that for all convex differentiable loss functions, $(w^\top\Phi)^\top\nabla_w R(w^\top\Phi)=0$ if and only if $w$ is optimal for all environments.

Although the penalty $\rho_e^{\mathrm{IRMv1}}(\varphi,1.0)$ seems to be an appropriate invariance penalty, which is easily implementable, its effectiveness to capture invariance has been brought into question. In particular, Rosenfeld et al. (2021) construct a non-invariant data representation such that the penalty $\rho_e^{\mathrm{IRMv1}}(\varphi,1.0)$ is arbitrarily small. In what follows, we propose an alternative invariance penalty that is directly comparable to risk. Hence, if the invariance penalty of a classifier is small, then so is the difference between its risk and the risk under the optimal classifier.

---

[1]In the presence of known exogenous (environment dependent) variables, one can utilize anchor regression Rothenhäusler et al. (2021) that is conceptually related to the invariance penalization. Instead of an invariance penalty, anchor regression relies on the projection onto the span of the said variables.

## 3    IRMv2: AN ALTERNATIVE PENALTY

We revisit the structure of the risk in order to propose an alternative penalty. In particular, in the following Lemma, we provide the sub-optimality gap of risk under an arbitrary classifier in comparison to the optimal classifier.

**Lemma 1.** *Consider squared loss function. Let $w \in \mathbb{R}^{d_\varphi}$ and $w_e^\star(\varphi)$ as defined in equation 4. Then,*

$$R_e\left(w^\top \varphi\right) = R_e\left(w_e^\star(\varphi)^\top \varphi\right) + \left\|\mathcal{I}_e(\varphi)^{1/2}\left(w - w_e^\star(\varphi)\right)\right\|^2. \tag{8}$$

Based on Lemma 1, we propose an invariance penalty that is directly comparable to risk.

$$\rho_e^{\mathrm{IRMv2}}(\varphi, w) := \left\|\mathcal{I}_e(\varphi)^{1/2}\left(w - w_e^\star(\varphi)\right)\right\|^2. \tag{9}$$

The relaxation of IRM using the penalty equation 9 is then given by

$$\min_{\varphi,\, w}\ \sum_{e \in \mathcal{E}_{\mathrm{tr}}} R_e(w^\top \varphi) + \lambda \rho_e^{\mathrm{IRMv2}}(\varphi, w). \tag{10}$$

We further simplify the relaxation equation 10 by finding its optimal classifier for a fixed data representation defined as

$$w^\star(\varphi) := \mathrm{argmin}_w\ \sum_{e \in \mathcal{E}_{\mathrm{tr}}} R_e\left(w^\top \varphi\right) + \lambda \rho_e^{\mathrm{IRMv2}}(\varphi, w). \tag{11}$$

In the following Lemma, we leverage on the structure of the squared loss to find $w^\star(\varphi)$.

**Lemma 2.** *Consider squared loss function and fix $\varphi$. Let $w_e^\star(\varphi)$ and $w^\star(\varphi)$ as defined in equation 4 and equation 11, respectively. Then,*

$$w^\star(\varphi) = \left(\sum_{e \in \mathcal{E}_{\mathrm{tr}}} \mathcal{I}_e(\varphi)\right)^{-1}\left(\sum_{e \in \mathcal{E}_{\mathrm{tr}}} \mathcal{I}_e(\varphi) w_e^\star(\varphi)\right). \tag{12}$$

*Moreover, it holds that*

$$\mathrm{argmin}_w\ \sum_{e \in \mathcal{E}_{\mathrm{tr}}} R_e\left(w^\top \varphi\right) = w^\star(\varphi). \tag{13}$$

Based on Lemmas 1 and 2, we propose the following relaxation of IRM, which we refer to by IRMv2.

$$\min_{\varphi}\ \sum_{e \in \mathcal{E}_{\mathrm{tr}}} R_e\left(w^\star(\varphi)^\top \varphi\right) + \lambda \rho_e^{\mathrm{IRMv2}}(\varphi, w^\star(\varphi)). \tag{IRMv2}$$

We provide the pseudo-code for IRMv2 in Algorithm 1 in Appendix A.

There are several comments in order. It is worth noting that Equation equation 13 in Lemma 1 reveals that for a fixed data representation, the optimal classifiers of ERM and IRMv2 are equal. However, this equality would not imply the equality of the ERM and IRMv2 predictors (i.e., the composition of the classifier and the data representation). The latter is due to the presence of the invariance penalty in the optimization of $\varphi$ in IRMv2 in comparison to ERM.

There are a number of factors distinguishing IRMv2 from IRMv1. First, IRMv2 relies on the optimal classifier $w^\star(\varphi)$ while $w = 1.0$ in IRMv1. Second, the loss function in IRMv2 is squared loss while IRMv1 allows for utilization of other loss functions. Although this additional flexibility of IRMv1 may seem appealing, the counterexample of Rosenfeld et al. (2021) shows the failure of the penalty of IRMv1 to capture invariance for logistic loss. More importantly, for squared loss, $\mathcal{I}_e(\varphi)$ is incorporated differently in the invariance penalty of IRMv1 and IRMv2. We formalize this latter observation in the Section 4.3.1.

### 3.1 IRMv1A: An Adaptive Penalty Coefficient

We first bound the invariance penalty of IRMv1 in terms of the penalty of IRMv2 and the eigenvalues of $\mathcal{I}_e(\varphi)$. Then, based on this comparison, we propose an adaptive approach in choosing the penalty coefficient for IRMv1, which we refer to as IRMv1-Adaptive (IRMv1A).

**Lemma 3.** *Let $\rho_e^{\mathrm{IRMv1}}(\varphi, w)$ and $\rho_e^{\mathrm{IRMv2}}(\varphi, w)$ be the invariance penalties of the IRMv1 and IRMv2 defined in Equations equation 6 and equation 9, respectively. Then,*

$$\lambda_{\min}(\mathcal{I}_e(\varphi)) \rho_e^{\mathrm{IRMv2}}(\varphi, w) \leq \rho_e^{\mathrm{IRMv1}}(\varphi, w) \leq \lambda_{\max}(\mathcal{I}_e(\varphi)) \rho_e^{\mathrm{IRMv2}}(\varphi, w). \tag{14}$$

The proof of Lemma 3 directly follows from the definition of the invariance penalties $\rho_e^{\mathrm{IRMv1}}(\varphi, w)$ and $\rho_e^{\mathrm{IRMv2}}(\varphi, w)$, and the fact that for a symmetric matrix $A \in \mathbb{R}^{d \times d}$ and a vector $u \in \mathbb{R}^d$, it holds that $\lambda_{\min}(A)\|u\|^2 \leq u^\top A u \leq \lambda_{\max}(A)\|u\|^2$.

As IRMv1 is used in practice in conjunction with losses other than squared loss for classification, in order to justify our adaptive penalty coefficient, we bound the difference of risk for cross-entropy loss under two different classifiers in terms of the invariance penalty of IRMv2.

**Lemma 4.** *Consider binary classification with cross-entropy loss. Then, for $w_1, w_2 \in \mathbb{R}^{d_\varphi}$ it holds that*

$$|R_e(w_1^\top \varphi) - R_e(w_2^\top \varphi)| \leq \left\| \mathcal{I}_e(\varphi)^{1/2}(w_1 - w_2) \right\|.$$

Using Lemmas 3 and 4, we suggest the following rule for the penalty coefficient of IRMv1.

$$\lambda_e := \frac{1}{\lambda_0 + \lambda_{\min}(\mathcal{I}_e(\varphi))}, \tag{15}$$

where $\lambda_0 \geq 0$ is a user-specified parameter. The role of $\lambda_0$ is to avoid numerical instability when $\lambda_{\min}(\mathcal{I}_e(\varphi))$ is small. Note that this is an adaptive rule, as $\varphi$ may change throughout training.

## 4 Theoretical Results and Analysis

In this section, we investigate the effectiveness of IRM and its practical implementations in capturing invariance by focusing on a setting in which the data is generated according to a Structural Equation Model (SEM) (Pearl, 2009). SEM refers to a set of equations specifying the relationship between variables, which is a common structural assumption in causal inference. In particular, under an SEM, variables are related through a causal graph such that each variable is only a function of its parents and an exogenous random variable. Similar to Arjovsky et al. (2019) and Rosenfeld et al. (2021), we establish conditions under which IRM recovers an invariant predictor for data generated according to SEMs with linear functions. We then investigate the role of the eigenstructure of $\mathcal{I}_e(\varphi)$, in particular, in relation to counterexamples of Arjovsky et al. (2019) and Rosenfeld et al. (2021), and provide practical solution to alleviate the possibility of such failure modes.

### 4.1 SEM for Classification

For each environment $e$, data $(X^e, Y^e)$ is generated according to the following SEM:

$$X^e = S \begin{bmatrix} Z_c \\ Z_e \end{bmatrix}, \quad Y^e = \begin{cases} 1, & \text{with prob. } \eta, \\ -1, & \text{with prob. } 1 - \eta, \end{cases} \tag{16}$$

where $\eta \in [0, 1]$, and $S \in \mathbb{R}^{d \times (d_e + d_c)}$ is a left invertible matrix, i.e., there exists $S^\dagger$ such that $S^\dagger S = I$. In this model, $Z_c$ captures the causal variables that are invariant across environments, and $Z_e$ captures the spurious environment dependent variables.

The variables $Z_c$ and $Z_e$ are generated as follows

$$Z_c = \mu_c Y + W_c, \quad W_c \sim \mathcal{N}(0, \sigma_c^2 I), \tag{17}$$

$$Z_e = \mu_e Y + W_e, \quad W_e \sim \mathcal{N}(0, \sigma_e^2 I). \tag{18}$$

Here, $\mu_c \in \mathbb{R}^{d_c}$, $\mu_e \in \mathbb{R}^{d_e}$, and $\mathcal{N}(\mu, \Sigma)$ denotes multi-variate Gaussian distribution with mean $\mu$ and covariance $\Sigma$. We further assume that $W_c$, $W_e$, and $Y^e$ are independent for all environments.

## 4.2 INVARIANT REPRESENTATION UNDER IRM

For the setting introduced in 4.1, the invariant data representation is linear. In particular, for any $d \geq d_c$, $\varphi(X^e) = \begin{bmatrix} I_{d_c} & \mathbf{0} \\ \mathbf{0} & \mathbf{0} \end{bmatrix} S^\dagger X^e = Z_c$ is an invariant data representation. Given that $X$ is also linear in $Y$, the class of linear representations may be sufficiently reach to elicit an invariant predictor. Naturally, the possibility of finding an invariant predictor depends on the number and the diversity of training environments. We now introduce non-degeneracy conditions on training environments under which IRM is guaranteed to find an invariant predictor, provided sufficient number of training environments.

Let $|\mathcal{E}_{\mathrm{tr}}| > d_e$. As $\mathrm{span}(\{\mu_e\}_{e \in \mathcal{E}_{\mathrm{tr}}}) \leq d_e$, for each $e \in \mathcal{E}_{\mathrm{tr}}$ there exists a set of coefficients $\alpha_i^e$ for $i \in \mathcal{E}_{\mathrm{tr}} \backslash e$ such that

$$\mu_e = \sum_{i \in \mathcal{E}_{\mathrm{tr}} \backslash e} \alpha_i^e \mu_i. \tag{19}$$

We say that $\mathcal{E}_{\mathrm{tr}}$ is a *non-degenerate set of environments* if for all $e \in \mathcal{E}_{\mathrm{tr}}$ it holds that

$$\sum_{i \in \mathcal{E}_{\mathrm{tr}} \backslash e} \alpha_i^e \neq 1, \tag{20}$$

$$\mathrm{rank}\left(\Gamma_e\right) = d_e, \tag{21}$$

where $\Gamma_e$ is defined as

$$\Gamma_e := \frac{1}{1 - \sum_{i \in \mathcal{E}_{\mathrm{tr}} \backslash e} \alpha_i^e} \left( \sigma_e^2 I + \mu_e \mu_e^\top - \sum_{i \in \mathcal{E}_{\mathrm{tr}} \backslash e} (\sigma_i^2 I + \mu_i \mu_i^\top) \alpha_i^e \right).$$

The conditions of equation 20 and equation 21 specify that the span of covariance matrices of $Z_e$'s is $R^{d_e}$. This is a natural requirement to eliminate the degrees of freedom on the dependency of the data representation on the environment dependent features. We note that the non-degeneracy conditions considered in (Rosenfeld et al., 2021) are similar to the ones introduced here, with the difference that instead of depending on covariance matrices of $Z_e$ as in equation 21, their condition relies only on the variances $\sigma_e^2$. This difference in the non-degeneracy requirements is due to the fact that they consider logistic loss and we consider squared loss.

**Theorem 1.** *Assume that $|\mathcal{E}_{\mathrm{tr}}| > d_e$ where $(X^e, Y^e)$ generated according to equation 16. Consider a linear data representation $\Phi X = A Z_c + B Z_e$ and a classifier $w(\Phi)$ on top of $\Phi$ that is invariant, i.e., $w(\Phi) = w_e^\star(\Phi)$ for all $e \in \mathcal{E}_{\mathrm{tr}}$. If non-degeneracy conditions Eqs. (19-21) holds, then either $w(\Phi) = 0$ or $B = 0$.*

Theorem 1 characterizes the connection between invariant predictors and the diversity of training environments in the linear setting. Its conclusions are directly applicable to IRM and its relaxations including IRMv1, v1A, and v2.

## 4.3 THE ROLE OF $\mathcal{I}_e(\varphi)$

Recall that the main difference between the invariance penalty of IRMv1 (with squared loss) and IRMv2 is the way that $\mathcal{I}_e(\varphi)$ is incorporated. Although at a first glance this may not seem significant, in what follows we demonstrate the importance of $\mathcal{I}_e(\varphi)$.

### 4.3.1 THE POTENTIAL FAILURE OF INVARIANCE PENALIZATION

Arjovsky et al. (2019) consider a linear Structural Equation Model (SEM) of the following form.

$$X_1 \sim \mathcal{N}(0, \sigma^2), \ Y = X_1 + Z_1 \text{ with } Z_1 \sim \mathcal{N}(0, \sigma^2), \ X_2 = Y + Z_2 \text{ with } Z_2 \sim (0, 1),$$

and $X = [X_1, X_2]^\top$. The variance $\sigma^2$ is the only parameter that changes across environments, and we assume that $Z_1$, $Z_2$, and $X_1$ are independent in and across all environments. Hence, in this setting, $X_1$ models the invariant features and $X_2$ models the environmental features as the correlation between $X_2$ and $Y$ varies across environments.

Consider the data representation $\varphi_c(x)$ parameterized by a variable $c \in \mathbb{R}$ as

$$\varphi_c(X) = \begin{bmatrix} X_1 \\ cX_2 \end{bmatrix}, \tag{22}$$

where $c = 0$ attains the invariant data representation, i.e., $\varphi_0(X) = [X_1, 0]^\top$. Based on this model, Arjovsky et al. (2019) argue that $\|w_{\text{inv}} - w_e^\star(\varphi_c)\|^2$ is a poor choice for the invariance penalty as it is discontinuous at the invariant representation with $c = 0$, and vanishes as $c \to \infty$. Interestingly, $\mathcal{I}_e(\varphi_c)$ is ill-conditioned for both small and large $c$'s. More precisely, in Appendix D.1 we show that

$$\lim_{c \to 0} \kappa(\mathcal{I}_e(\varphi_c)) = \lim_{c \to +\infty} \kappa(\mathcal{I}_e(\varphi_c)) = +\infty,$$

where $\kappa(\cdot)$ denotes the condition number. That is, for a normal matrix $A$, its condition number is $\kappa(A) := |\lambda_{\max}(A)|/|\lambda_{\min}(A)|$ where $\lambda_{\max}$ and $\lambda_{\min}$ denote its maximum and minimum eigenvalues, respectively.

We now examine the counterexample introduced by Rosenfeld et al. (2021), which is based on the SEM setting introduced in Section 4.1 and equation 16. They consider the data representation $\varphi_\epsilon(X^e)$ defined as

$$\varphi_\epsilon(X^e) := \begin{bmatrix} Z_c \\ 0 \end{bmatrix} + \begin{bmatrix} 0 \\ Z_e \end{bmatrix} \mathbb{1}_{\{Z_e \notin \mathcal{Z}_\epsilon\}}, \tag{23}$$

where $\{Z_e \notin \mathcal{Z}_\epsilon\}$ is an event with $\mathbf{P}(Z_e \in \mathcal{Z}_\epsilon) \leq p_{e,\epsilon}$ where $p_{e,\epsilon} := \exp(-d_e \min\{\epsilon - 1, (\epsilon - 1)^2\}/8)$. The parameter $\epsilon \geq 1$ control the degrees to which the environmental variable $Z_e$ appears in the data representation by controlling the set $\mathcal{Z}_\epsilon$ (see equation 40 in the Appendix). In particular, $\varphi_\epsilon(X^e)$ converges to the invariant representation as $\epsilon \to \infty$, i.e., $\lim_{\epsilon \to \infty} \varphi_\epsilon(X^e) = \varphi_{inv}(X^e) := [Z_c, 0]^\top$.

Rosenfeld et al. (2021) show that IRMv1's invariance penalty $\|\nabla_w R_e(w^\top \varphi)\|^2 = O(p_{e,\epsilon}^2)$, and can be made arbitrarily small, hence it is poor discrepancy as an invariance penalty. This, together with the fact that the training risk under $\varphi_\epsilon$ is close to the risk under the invariant representation, they argue that $\varphi_\epsilon$ is a plausible representation under IRMv1. However, this observation is only true for the case where $\lambda$ is a constant. More specifically, $\varphi_\epsilon$ is not a plausible solution of IRMv1 with adaptive penalty coefficient, e.g., $\lambda(\varphi) = 1/\lambda_{\min}(\mathcal{I}_e(\varphi))^2$.

### 4.3.2 PRACTICAL IMPLICATIONS

In both of the examples discussed in Section 4.3.1, $\mathcal{I}(\varphi) = \mathbf{E}[\varphi(X)\varphi(X)^\top]$ is singular for the invariant representation. This is due to the fact that for both of the examples, the span of data representations $\varphi(X)$ is the same as the span of inputs $X$ while the span of the invariant features is only a small subset of that space. As a consequence, in presence of environment dependent variables, one should avoid utilizing data representations with codomain that is the same as the input space. In particular, when $\varphi$ is parameterized by a neural network, the common approach of setting the size of the last layer equal to $K - 1$ for $K$-class classification, and 1 for regression problem avoids such phenomenon. We further illustrate such behavior on the Colored MNIST experiment, a standard benchmark for evaluating the performance of domain generalization methods.

Figure 1 illustrate that the test accuracy of IRMv1 drastically drops as we increase the dimension of the representation. However, the test accuracy of IRMv2 gradually increases first before dropping in a considerably overflexible representation. It is worth noting that the condition number of $\mathcal{I}(\varphi)$ increases with $d_\varphi$ and is ill-conditioned for large $d_\varphi$, e.g., $\kappa(\mathcal{I}(\varphi)) > 10^{12}$ when $d_\varphi = 2048$ (see Figure 2 in the Appendix). Note that in this experiment, images are down-sampled and contain $14 \times 14$ pixels, i.e., $d_X = 196$.

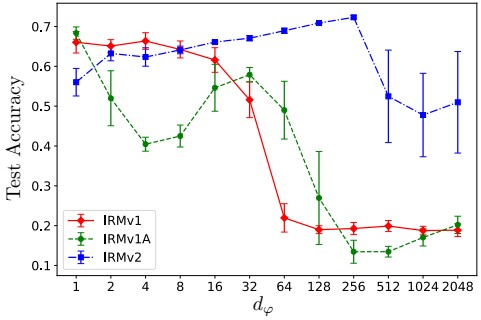

Figure 1: Effect of representation dimension on test accuracy for IRMv1, IRMv1A, and IRMv2.

## 5 EXPERIMENTS

In this section, we empirically evaluate the efficacy of our proposed implementations of IRM, namely, IRMv2 and IRMv1A with IRMv1 and ERM. We demonstrate the appeal of our approach on *InvarianceUnitTests* (Aubin et al., 2020), a synthetic test bed for evaluation of OoD methods with multiple experiments and data sets, each of which embedding a different causal relationship. We then evaluate each method on our curated data set *HealthyGutTests* to illustrate how the principle of invariant risk minimization can be used to minimize disparities and discrepancies in biomedical research. For completeness, we also perform numerical experiments on *DomainBed*, an extensive framework released by (Gulrajani & Lopez-Paz, 2020) to test domain generalization algorithms for image classification tasks on various benchmark data sets. Similar to Gulrajani & Lopez-Paz (2020), we also observe that none of the domain generalization algorithms significantly outperform ERM. We refer the reader to Appendix E for the detailed results and further discussion.

### 5.1 INVARIANCEUNITTESTS

In this section, we evaluate the efficacy of our proposed approaches for invariance discovery on the *InvarianceUnitTests* recently proposed by Aubin et al. (2020). These unit-tests entail three classes of low-dimensional linear problems, each capturing a different structure for inducing spurious correlations. The data set for each problem falls within the multi-environment setting described in Section 2 with $n_e = 10^4$. For all problems, the input $x^e \in \mathbb{R}^d$ is constructed as $x^e = (x^e_{\text{inv}}, x^e_{\text{spu}})$ where $x^e_{\text{inv}} \in \mathbb{R}^{d_{\text{inv}}}$ and $x^e_{\text{spu}} \in \mathbb{R}^{d_{\text{spu}}}$ denote the invariant and the spurious features, respectively. To make the problems more realistic, Aubin et al. (2020) repeat each experiment and *scramble* the inputs by multiplying $x^e$ by a rotation matrix. In each problem, the spurious correlations that exist in the training environments are discarded in the test environment by random shuffling. As a basis for comparison, similar to Aubin et al. (2020), we implement an Oracle ERM where the spurious correlations are shuffled in the training data sets as well, and hence, ERM can readily identify them.

Table 1 illustrates the training and test errors of the various methods on linear unit test experiments. One can observe that IRMv2 is the only method that discovered a nearly invariant predictor on all experiments. However, on classification tasks (Example 2), all other methods except IRMv1A attained near perfect training accuracy by relying on the spurious correlations while their test accuracy was comparable to random guessing as the spurious features were randomly shuffled in the test set. The stability of IRMv2 in discovering invariant predictors makes it a suitable fit for real-world applications as we will demonstrate in the following section.

|  | ERM | | IRMv1 | | IRMv1A | | IRMv2 | | Oracle | |
|---|---|---|---|---|---|---|---|---|---|---|
|  | Training | Test | Training | Test | Training | Test | Training | Test | Training | Test |
| Example1 | 7.35 | 14.40 | 9.47 | **10.72** | 12.82 | 12.87 | 11.14 | 11.26 | 10.49 | 10.32 |
| Example1s | 7.32 | 14.37 | 9.36 | **10.76** | 12.76 | 13.24 | 11.89 | 12.25 | 10.43 | 10.53 |
| Example2 | 0.03 | 0.42 | 0.03 | 0.45 | 0.11 | **0.31** | 0.26 | 0.34 | 0.00 | 0.00 |
| Example2s | 0.03 | 0.45 | 0.03 | 0.45 | 0.04 | **0.27** | 0.22 | 0.38 | 0.00 | 0.00 |
| Example3 | 0.00 | 0.37 | 0.00 | 0.36 | 0.49 | **0.26** | 0.31 | 0.40 | 0.01 | 0.01 |
| Example3s | 0.00 | 0.37 | 0.00 | 0.37 | 0.49 | **0.34** | 0.30 | 0.41 | 0.01 | 0.01 |

Table 1: Training and Test Errors on *InvarianceUnitTests* for all algorithms and examples with $(d_{\text{inv}}, d_{\text{spu}}, d_{\text{env}}) = (5, 5, 3)$. The errors for Examples 1 and 1s are in MSE and all others are classification error. The empirical mean and the standard deviation are computed using 10 independent experiments. An 's' indicates the scrambled variation of its corresponding problem setting.

### 5.2 HEALTHYGUTTESTS

In this section, we further evaluate the efficacy of our proposed approaches on a set of experiments that highlight the importance of generalizability in real-world applications. An increasing number of microbiome studies have associated various alterations in human gut microbiota composition with both chronic and acute diseases, providing strong evidence that the gut microbiome is an essential factor in the maintenance of human health. *HealthyGutTests* is a curated set of microbiomes extracted from an extensively phenotyped and standardized meta-analysis cohort (Gupta et al., 2020) of 1770 individuals across 10 independent studies (Supplementary Table 7). Study selection was

limited to case-control studies in adult populations wherein Whole-Genome Sequencing (WGS) metagenome data of human stool (gut microbiome) and corresponding subject meta-data were publicly available. To reduce class imbalance, studies consisting of only control samples (healthy) or those with fewer than 100 total samples were excluded.

This data set reflects the characteristics of the domain generalization problem setting, as the data for each study is collected by different researchers, across various timepoints, locations and demographic populations. With this gut microbiome data set, we seek to illustrate a use-case in which the IRM principle can be used to minimize the effect of spurious correlations confounding the discovery of real explanatory variables and thus build a fair and trustworthy ML model that is appropriate for real-world deployment in a biomedical setting. We consider the task of *human health status classification*: predicting the binary label (healthy/unhealthy) associated with a sample. The data set falls within the multi-environment setting described in Section 2, with $n_e = 10$ and inputs $x^e$ consisting of 992 features, where each feature corresponds to a unique microbial species. When data from multiple studies are collectively analyzed, systematic variation across different studies can give rise to biases known as "batch-effects". These batch effects may obscure biologically relevant and under-represented sub-population effects in the data that are difficult to identify. When viewed in a clinical setting, these sub-population effects may represent differential etiologies, and thus require different treatment strategies. An ERM classifier trained on all such data (i.e. by shuffling and treating data as iid) may learn these batch effects as predictors of health status, and consequently ignore invariant features that are indicative of under-represented subpopulation effects. We seek to build an invariant predictor that is robust in its predictions when trained and tested on different experimental splits across the data set's ten studies. Table 2 illustrates the test accuracy of the various methods on this binary classification task. One can observe that IRMv2 and IRMv1A methods demonstrate superior performance when compared to ERM and perform as good if not better than IRMv1 across 9 out of 10 test splits.

|  | ERM | IRMv1 | IRMv1A | IRMv2 |
|---|---|---|---|---|
| Feng (2015) | 0.785 | 0.856 | 0.848 | **0.856** |
| He (2017) | 0.604 | 0.604 | 0.604 | **0.606** |
| Jie (2017) | 0.733 | 0.726 | 0.724 | **0.734** |
| Karlsson (2013) | 0.813 | 0.861 | 0.849 | **0.866** |
| Le Chatelier (2013) | 0.642 | **0.655** | 0.653 | 0.653 |
| Nielsen (2014) | 0.732 | 0.744 | 0.739 | **0.744** |
| Qin (2012) | 0.763 | 0.785 | 0.779 | **0.795** |
| Vogtmann (2016) | 0.692 | 0.696 | **0.700** | **0.700** |
| Zeller (2014) | 0.775 | 0.755 | 0.768 | **0.787** |
| Zhang (2015) | 0.666 | 0.700 | **0.701** | **0.701** |
| Average | $0.721 \pm 0.064$ | $0.738 \pm 0.077$ | $0.736 \pm 0.074$ | $\mathbf{0.744 \pm 0.079}$ |

Table 2: Test Accuracy on *HealthyGutTests* health status classification task. Each test accuracy is averaged over 5 independent experiments.

## 6 CONCLUSION

In this paper, we have presented IRMv2, an alternative implementation of the IRM principle that aims to enable out-of-distribution generalization by finding environment invariant predictors. We establish theoretical results on the effectiveness of our approach in the linear setting. In doing so, we bring forward the importance of the eigenstructure of the Gramian matrix of the data representation. In particular, we show that for the existing counterexample on the potential failure of IRMv1, the aforementioned matrix is ill-conditioned for the invariant representation. This highlights the significance of the span of the data representations in relation to the span of the underlying true invariant features of the data. That is, if the data representation allows for more degrees of freedom than needed to capture invariance, the Gramian matrix of the invariant representation would be ill-conditioned. This observation provides intuition on the underlying reasons why current implementations of IRM may fail. Our work demonstrates the effectiveness of IRMv2 and IRMv1A on both synthetic and real-world data sets. While this investigation of the IRM principle has lead to the development of a more pragmatic solution, we leave for future work, establishing theoretical guarantees beyond linear regimes.

## REPRODUCABILITY STATEMENT

We have provided detailed mathematical proofs of all of our theoretical claims in the appendices. The data processing, neural network architecture, and the model selection for Colored MNIST, InvarianceUnitTests, and DomainBed experiments are exactly replicated from their respective author repositories. For HealthyGutTests experiments, we utilized the data sets curated and standardized by domain experts Gupta et al. (2020). We discuss our additional data processing step in Section 5.2. We also provide details of the neural network architecture and the model selection in Appendix F.

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

## A  IRMv2 ALGORITHM PSEUDO-CODE

---

**Algorithm 1** IRMv2

---

1: **Input:** Data set: $D_e$ for $e \in \mathcal{E}_{\mathrm{tr}}$. Loss function: Squared loss, Parameters: penalty coefficient $\lambda \geq 0$, data representation parameters $\theta \in \mathbb{R}^{d_\theta}$, learning rate $\eta_t$, training horizon $T$.

2: **Initialize** $\theta_1$ randomly

3: **for** $t = 1, 2, \ldots, T$ **do**
4:    **for** $e \in \mathcal{E}_{\mathrm{tr}}$ **do**
5:       compute the LSE $w_e^\star(\varphi_{\theta_t})$ according to Eq. equation 4
6:    compute the optimal classifier $w^\star(\varphi_{\theta_t})$ according to Eq. equation 12
7:    $\mathcal{L}_t(\varphi_{\theta_t}) \leftarrow \sum_{e \in \mathcal{E}_{\mathrm{tr}}} \mathcal{R}_e(w^\star(\varphi_{\theta_t})^\top \varphi_{\theta_t}) + \lambda \rho_e^{\mathrm{IRMv2}}(\varphi_{\theta_t}, w^\star(\varphi_{\theta_t}))$
8:    $\theta_{t+1} \leftarrow \theta_t - \eta_t \nabla_{\theta_t} \mathcal{L}_t(\varphi_{\theta_t})$
9: **Output** predictor $w^\star(\varphi_{\theta_T})^\top \varphi_{\theta_T}$.

---

## B  CONSIDERATIONS FOR PRACTICAL IMPLEMENTATION

As the data generating distributions for each environment are unknown, we provide further details on practical implementations of IRMv2. As training is often done by gradient descent on mini-batches, we focus on this setting. For simplicity of notation, assume that the sample size of all environments are equal, each of which is divided into $b$ mini-batches of size $m$. Then, $\widehat{w}_e^\star$ the empirical estimate of the optimal classifier $w_e^\star$ is given by

$$\widehat{w}_e^\star(\varphi) := \left( \sum_{k=1}^b \Phi_k^{e\top} \Phi_k^e \right)^{-1} \sum_{k=1}^b \Phi_k^{e\top} Y_k^e, \tag{24}$$

where $\Phi_k^e \in \mathbb{R}^{m \times d_\varphi}$ is a matrix each of its row is $\varphi(x_i^e)^\top$ of the $k$-th mini-batch of environment $e$, and with some abuse of notation $Y_k^e \in \mathbb{R}^m$ is defined as the vector containing $y_e^i$ for the same mini-batch.

**Bias:** Note that whether the estimator equation 24 is unbiased or not depends on the underlying data generating assumption. In particular, it may be natural to assume that for the underlying invariant representation, it holds that $\mathbf{E}[Y^e | \varphi_{\mathrm{inv}}(X^e)] = w_{\mathrm{inv}}^\top \varphi_{\mathrm{inv}}(X^e)$ with the error $Y^e - \mathbf{E}[Y^e | \varphi_{\mathrm{inv}}(X^e)]$ being independent of $X^e$ and $Y^e$, e.g., $y_e^i = w_{\mathrm{inv}}^\top \varphi_{\mathrm{inv}}(x_i^e) + \eta_i$ where $\{\eta\}_{i=1}^n$ is a sequence of zero-mean i.i.d. random variables. However, this may not hold for an arbitrary representation $\varphi$. One can modify the estimator equation 24 to derive an unbiased estimator by using different mini-batches for each term, e.g., by randomly dividing the mini-batches into two parts. More specifically, let $B$ be the set of randomly chosen mini-batches of size $\lfloor b/2 \rfloor$, and $[b] \backslash B$ be the rest of the mini-batches. Let $\mathcal{B}$ be a set of such $B$s. Then,

$$\widehat{w}_e^\star(\varphi) = \frac{1}{|\mathcal{B}|} \sum_{B \in \mathcal{B}} \left\{ \left( \sum_{k \in B} \Phi_k^{e\top} \Phi_k^e \right)^{-1} \sum_{k \in [b] \backslash B} \Phi_k^{e\top} Y_k^e \right\}. \tag{25}$$

The estimator equation 25 is unbiased if $x_i^e$ are i.i.d. and $y_i^e$ is independent of all other $x_j^e$ conditioned on $x_i^e$.

**Numerical Stability:** Although $\sum_{k \in B} \Phi_k^{e\top} \Phi_k^e$ is positive semi-definite almost surely, it is singular for $m|B| < d_\varphi$ and may even be singular for $m|B| > d_\varphi$. Hence, to improve numerical stability, we consider the following empirical estimate of $\mathcal{I}_e(\varphi)$

$$\widehat{\mathcal{I}}_{e,\lambda,B}(\varphi) := \lambda I + \sum_{k \in B} \Phi_k^{e\top} \Phi_k^e,$$

where $\lambda \geq 0$. That is, to improve the numerical stability, we consider the $\ell_2$ regularized least squares estimate[2]

$$\widehat{w}_{e,B}^{\star}(\varphi) = \widehat{\mathcal{I}}_{e,\lambda,B}(\varphi)^{-1} \sum_{k \in [b] \setminus B} \Phi_k^{e\top} Y_k^e. \tag{26}$$

Then,

$$\widehat{w}_B^{\star}(\varphi) = \left( \sum_{e \in \mathcal{E}_{\text{tr}}} \widehat{\mathcal{I}}_{e,\lambda,B}(\varphi) \right)^{-1} \sum_{e \in \mathcal{E}_{\text{tr}}} \widehat{\mathcal{I}}_{e,\lambda,B}(\varphi) \widehat{w}_{e,B}^{\star}(\varphi).$$

We now provide an empirical estimate of the invariant penalty $\rho_e^{\text{IRMv2}}$. We further partition $B$ into $B_1, B_2$, and $B_3$. Then,

$$\widehat{\rho}_e^{\text{IRMv2}} = \frac{1}{|\mathcal{B}|} \sum_{B \in \mathcal{B}} \left\{ \left( \widehat{w}_{e,B_1}^{\star}(\varphi) - \widehat{w}_{B_1}^{\star}(\varphi) \right)^{\top} \widehat{\mathcal{I}}_{e,\lambda,B_2}(\varphi) \left( \widehat{w}_{e,B_3}^{\star}(\varphi) - \widehat{w}_{B_3}^{\star}(\varphi) \right) \right\}.$$

Note that $\widehat{\rho}_e^{\text{IRMv2}}$ is an unbiased estimator of $\rho_e^{\text{IRMv2}}$ for $\lambda = 0$.

## C  MATHEMATICAL PROOFS

### C.1  PROOF OF LEMMA 1

First, the risk under environment $e$ with $w = w_e^{\star}(\varphi)$ is given by

$$R_e \left( w_e^{\star}(\varphi)^{\top} \varphi \right) = \mathbf{E}_{Y^e} \left[ \|Y^e\|^2 \right] - \mathbf{E}_{X^e, Y^e} \left[ \varphi(X^e) Y^e \right]^{\top} \mathcal{I}_e(\varphi)^{-1} \mathbf{E}_{X^e, Y^e} \left[ \varphi(X^e) Y^e \right].$$

Then,

$$R_e \left( w^{\top} \varphi \right) - R_e \left( w_e^{\star}(\varphi)^{\top} \varphi \right)$$
$$= w^{\top} \mathcal{I}_e(\varphi) w - 2 w^{\top} \mathbf{E}_{X^e, Y^e} \left[ \varphi(X^e) Y^e \right] + \mathbf{E}_{X^e, Y^e} \left[ \varphi(X^e) Y^e \right]^{\top} \mathcal{I}_e(\varphi)^{-1} \mathbf{E}_{X^e, Y^e} \left[ \varphi(X^e) Y^e \right]$$
$$= \left\| \mathcal{I}_e(\varphi)^{1/2} \left( w - w_e^{\star}(\varphi) \right) \right\|^2.$$

### C.2  PROOF OF LEMMA 2

Recall the definition of $w^{\star}(\varphi)$

$$w^{\star}(\varphi) = \operatorname{argmin}_w \sum_{e \in \mathcal{E}_{\text{tr}}} R_e \left( w^{\top} \varphi \right) + \lambda \rho_e^{\text{IRMv2}}(\varphi, w).$$

That is,

$$w^{\star}(\varphi) = \operatorname{argmin}_w \sum_{e \in \mathcal{E}_{\text{tr}}} \mathbf{E} \left[ \left\| w^{\top} \varphi(X^e) - Y^e \right\|^2 \right] + \lambda \left\| \mathcal{I}_e(\varphi)^{1/2} \left( w - w_e^{\star}(\varphi) \right) \right\|^2$$
$$= \operatorname{argmin}_w \sum_{e \in \mathcal{E}_{\text{tr}}} \left\{ (1 + \lambda) w^{\top} \mathcal{I}_e(\varphi) w - 2 w^{\top} \left( \mathbf{E} \left[ \varphi(X^e) Y^e \right] + \lambda \mathcal{I}_e(\varphi) w_e^{\star}(\varphi) \right) \right.$$
$$\left. + w_e^{\star}(\varphi)^{\top} \mathcal{I}_e(\varphi) w_e^{\star}(\varphi) + \mathbf{E} \left[ \|Y^e\|^2 \right] \right\}.$$

---

[2]The $\ell_2$ regularized least squares estimate of $w_e^{\star}$ is defined as

$$\widehat{w}_e^{\star}(\varphi) = \operatorname{argmin}_w \sum_{k=1}^{b} \sum_{i=1}^{m} (y_{i,k}^e - w^{\top} \varphi(x_{i,k}^e))^2 + \lambda \|w\|_2^2.$$

Note that the objective function is a convex quadratic function of $w$. Hence, using the first-order optimality condition we have that

$$(1 + \lambda) \left( \sum_{e \in \mathcal{E}_{\mathrm{tr}}} \mathcal{I}_e(\varphi) \right) w^\star(\varphi) - \sum_{e \in \mathcal{E}_{\mathrm{tr}}} \left( \mathbf{E}\left[ \varphi(X^e) Y^e \right] + \lambda \mathcal{I}_e(\varphi) w_e^\star(\varphi) \right) = 0.$$

Then,

$$w^\star(\varphi) = \frac{1}{1 + \lambda} \left( \sum_{e \in \mathcal{E}_{\mathrm{tr}}} \mathcal{I}_e(\varphi) \right)^{-1} \left( \sum_{e \in \mathcal{E}_{\mathrm{tr}}} \left( \mathbf{E}\left[ \varphi(X^e) Y^e \right] + \lambda \mathcal{I}_e(\varphi) w_e^\star(\varphi) \right) \right).$$

Recall that $w_e^\star(\varphi) = \mathcal{I}_e(\varphi)^{-1} \left[ \varphi(X^e) Y^e \right]$. Then, $\left[ \varphi(X^e) Y^e \right] = \mathcal{I}_e(\varphi) w_e^\star(\varphi)$. Thus,

$$w^\star(\varphi) = \left( \sum_{e \in \mathcal{E}_{\mathrm{tr}}} \mathcal{I}_e(\varphi) \right)^{-1} \left( \sum_{e \in \mathcal{E}_{\mathrm{tr}}} \mathcal{I}_e(\varphi) w_e^\star(\varphi) \right).$$

Finally, using a similar argument, we get

$$\mathrm{argmin}_w \sum_{e \in \mathcal{E}_{\mathrm{tr}}} R_e \left( w^\top \varphi \right) = \mathrm{argmin}_w \sum_{e \in \mathcal{E}_{\mathrm{tr}}} \mathbf{E}\left[ \left\| w^\top \varphi(X^e) - Y^e \right\|^2 \right]$$

$$= \mathrm{argmin}_w \sum_{e \in \mathcal{E}_{\mathrm{tr}}} w^\top \mathcal{I}_e(\varphi) w - 2 w^\top \mathbf{E}\left[ \varphi(X^e) Y^e \right] + \mathbf{E}\left[ \left\| Y^e \right\|^2 \right]$$

$$= \left( \sum_{e \in \mathcal{E}_{\mathrm{tr}}} \mathcal{I}_e(\varphi) \right)^{-1} \left( \sum_{e \in \mathcal{E}_{\mathrm{tr}}} \mathcal{I}_e(\varphi) w_e^\star(\varphi) \right).$$

## C.3 PROOF OF LEMMA 3

For a symmetric matrix $A \in \mathbb{R}^{d \times d}$ and a vector $u \in \mathbb{R}^d$, it holds that $\lambda_{\min}(A) \|u\|^2 \leq u^\top A u \leq \lambda_{\max}(A) \|u\|^2$. Let $u = \mathcal{I}_e(\varphi)^{1/2} \left( w - w_e^\star(\varphi) \right)$ and $A = \mathcal{I}_e(\varphi)$. Then,

$$\left\| \mathcal{I}_e(\varphi) \left( w - w_e^\star(\varphi) \right) \right\|^2 \leq \lambda_{\max}(\mathcal{I}_e(\varphi)) \left\| \mathcal{I}_e(\varphi)^{1/2} \left( w - w_e^\star(\varphi) \right) \right\|^2$$

$$\left\| \mathcal{I}_e(\varphi) \left( w - w_e^\star(\varphi) \right) \right\|^2 \geq \lambda_{\min}(\mathcal{I}_e(\varphi)) \left\| \mathcal{I}_e(\varphi)^{1/2} \left( w - w_e^\star(\varphi) \right) \right\|^2.$$

## C.4 PROOF OF LEMMA 4

Consider binary classification where $Y \in \{0, 1\}$ with binary cross-entropy loss, i.e.,

$$\ell(f(X), Y) = -Y f(X) + \log(1 + \exp(f(X))).$$

Our objective is to bound the difference $|R(w_1^\top \varphi) - R(w_2^\top \varphi)|$ for two arbitrary classifiers $w_1$ and $w_2$.

Let $g(w) := \log(1 + \exp(w^\top \varphi(X)))$. Then, using the multivariate mean-value theorem, there exists $c \in (0, 1)$ such that

$$g(w_2) = g(w_1) + \nabla g((1 - c) w_1 + c w_2)^\top (w_2 - w_1).$$

We have that

$$\nabla g(w) = \frac{\exp(w^\top \varphi(X))}{1 + \exp(w^\top \varphi(X))} \varphi(X) = \sigma(w^\top \varphi(X)) \varphi(X),$$

where $\sigma(\cdot)$ denotes the sigmoid function. Let $\bar{w} := (1 - c) w_1 + c w_2$. Then,

$$\ell(w_2^\top \varphi(X), Y) - \ell(w_1^\top \varphi(X), Y)$$

$$= -Y w_2^\top \varphi(X) + \log(1 + \exp(w_2^\top \varphi(X))) + Y w_1^\top \varphi(X) - \log(1 + \exp(w_1^\top \varphi(X)))$$

$$= -Y (w_2 - w_1)^\top \varphi(X) + \sigma(\bar{w}^\top \varphi(X)) \varphi(X)^\top (w_2 - w_1)$$

$$= - \left( Y - \sigma(\bar{w}^\top \varphi(X)) \right) (w_2 - w_1)^\top \varphi(X),$$

where the second equality follows from the application of mean-value theorem. Then,

$$
\begin{aligned}
|R(w_1^\top \varphi) - R(w_2^\top \varphi)|^2 &= \left| \mathbf{E} \left[ \left( Y - \sigma(\bar{w}^\top \varphi(X)) \right) (w_2 - w_1)^\top \varphi(X) \right] \right|^2 \\
&\leq \mathbf{E} \left[ \left( Y - \sigma(\bar{w}^\top \varphi(X)) \right)^2 \right] \mathbf{E} \left[ \left( (w_2 - w_1)^\top \varphi(X) \right)^2 \right] \\
&\leq (w_2 - w_1)^\top \mathbf{E} \left[ \varphi(X) \varphi(X)^\top \right] (w_2 - w_1),
\end{aligned}
$$

where the first equality follows from Cauchy-Schwarz and the second one follows from the fact that $Y \in \{0, 1\}$ and $\sigma : \mathbb{R} \to [0, 1]$, hence $|Y - \sigma(\bar{w}^\top \varphi(X)| \leq 1$ almost surely.

### C.5  PROOF OF THEOREM 1

In order to prove Theorem 1, we first find the optimal classifier for a given data representation for the linear setting in the following Lemma.

**Lemma 5.** *Let $w_e^\star(\Phi) = \operatorname{argmin}_w R_e(w^\top \Phi)$ where $R_e(w^\top \Phi)$ is defined as*

$$
R_e(w^\top \Phi) = \mathbf{E} \left[ tr \left( \left( w^\top \Phi X^e - \begin{bmatrix} \mathbb{1}_{\{Y=+1\}} \\ \mathbb{1}_{\{Y=-1\}} \end{bmatrix} \right) \left( w^\top \Phi X^e - \begin{bmatrix} \mathbb{1}_{\{Y=+1\}} \\ \mathbb{1}_{\{Y=-1\}} \end{bmatrix} \right)^\top \right) \right].
$$

*Then,*

$$
w_e^\star(\Phi) = \left[ \eta \beta_e^\star(\Phi) \quad (1-\eta)\beta_e^\star(\Phi) \right],
$$

*where*

$$
\beta_e^\star(\Phi) = \frac{1}{1 + \bar{\mu}_e^\top \bar{\Sigma}_e^{-1} \bar{\mu}_e} \bar{\Sigma}_e^{-1} \bar{\mu}_e. \tag{27}
$$

*Here, $\bar{\Sigma}_e$ and $\bar{\mu}_e$ are defined as*

$$
\bar{\Sigma}_e := \Phi S \begin{bmatrix} \sigma_c^2 I & 0 \\ 0 & \sigma_e^2 I \end{bmatrix} S^\top \Phi^\top,
$$

$$
\bar{\mu}_e := \Phi S \begin{bmatrix} \mu_c \\ \mu_e \end{bmatrix}.
$$

*Proof of Lemma 5:* We have that

$$
w_e^\star(\Phi) = \mathcal{I}_e(\Phi)^{-1} \mathbf{E}_{X^e, Y^e} \left[ \Phi X^e \tilde{Y}^{e\top} \right].
$$

First, we have that

$$
\begin{aligned}
\mathbf{E}_{X^e, Y^e} \left[ \Phi X^e \tilde{Y}^{e\top} \right] &= \Phi S \mathbf{E}_{X^e, Y^e} \left[ \begin{bmatrix} Z_c \\ Z_e \end{bmatrix} \begin{bmatrix} \mathbb{1}\{Y=1\} \\ \mathbb{1}\{Y=-1\} \end{bmatrix}^\top \right] \\
&= \Phi S \begin{bmatrix} \eta \mu_c & (1-\eta)\mu_c \\ \eta \mu_e & (1-\eta)\mu_e \end{bmatrix} \\
&= \left[ \eta \bar{\mu}_e \quad (1-\eta)\bar{\mu}_e \right].
\end{aligned}
$$

Thus,

$$
w_e^\star(\Phi) = \left[ \eta \beta_e^\star(\Phi) \quad (1-\eta)\beta_e^\star(\Phi) \right], \tag{28}
$$

where

$$
\beta_e^\star(\Phi) = \mathcal{I}_e(\Phi)^{-1} \bar{\mu}_e. \tag{29}
$$

We now compute $\mathcal{I}_e(\Phi)^{-1}$ in terms of $\bar{\Sigma}_e^{-1}$ and $\bar{\mu}_e$. We have that

$$
\mathcal{I}_e(\Phi) = \mathbf{E}_{X^e} \left[ \Phi X^e X^{e\top} \Phi^\top \right] = \Phi S \mathbf{E} \left[ \begin{bmatrix} Z_c \\ Z_e \end{bmatrix} \begin{bmatrix} Z_c \\ Z_e \end{bmatrix}^\top \right] S^\top \Phi^\top.
$$

From the definition of $Z_c$ and $Z_e$, it follows that

$$\mathbf{E}\left[\begin{bmatrix} Z_c \\ Z_e \end{bmatrix} \begin{bmatrix} Z_c \\ Z_e \end{bmatrix}^\top\right] = \begin{bmatrix} \sigma_c^2 I & 0 \\ 0 & \sigma_e^2 I \end{bmatrix} + \begin{bmatrix} \mu_c \\ \mu_e \end{bmatrix} \begin{bmatrix} \mu_c \\ \mu_e \end{bmatrix}^\top.$$

Thus,

$$\mathcal{I}_e(\Phi) = \bar{\Sigma}_e + \bar{\mu}_e \bar{\mu}_e^\top$$

Using Sherman-Morrison formula, we have that

$$\mathcal{I}_e(\Phi)^{-1} = \bar{\Sigma}_e^{-1} - \frac{\bar{\Sigma}_e^{-1} \bar{\mu}_e \bar{\mu}_e^\top \bar{\Sigma}_e^{-1}}{1 + \bar{\mu}_e^\top \bar{\Sigma}_e^{-1} \bar{\mu}_e}. \tag{30}$$

Finally, using Equation equation 29 it follows that

$$\beta_e^\star(\Phi) = \left(\bar{\Sigma}_e^{-1} - \frac{\bar{\Sigma}_e^{-1} \bar{\mu}_e \bar{\mu}_e^\top \bar{\Sigma}_e^{-1}}{1 + \bar{\mu}_e^\top \bar{\Sigma}_e^{-1} \bar{\mu}_e}\right) \bar{\mu}_e = \frac{1}{1 + \bar{\mu}_e^\top \bar{\Sigma}_e^{-1} \bar{\mu}_e} \bar{\Sigma}_e^{-1} \bar{\mu}_e.$$

$\blacksquare$

The proof of Theorem 1 closely follows from the proof of Rosenfeld et al., 2021, Lemma C.3.. In what follows, we include a proof to keep the manuscript self-contained.

*Proof of Theorem 1:* First, notice that the decomposition $\varphi(X^e) = AZ_c + BZ_e$ (or $\Phi S = [A \quad B]$) is without loss of generality under the assumption of left-invertibilibity of $S$. Then,

$$\bar{\Sigma}_e = \sigma_c^2 A A^\top + \sigma_e^2 B B^\top, \tag{31}$$
$$\bar{\mu}_e = A\mu_c + B\mu_e. \tag{32}$$

Recall from Lemma 5 that $\beta_e^\star(\Phi) = \bar{\Sigma}_e^{-1} \bar{\mu}_e / (1 + \bar{\mu}_e^\top \bar{\Sigma}_e^{-1} \bar{\mu}_e)$. If $\beta_e^\star(\Phi)$ is invariant, then $\beta^\star = \beta_e^\star(\Phi)$ for all $e \in \mathcal{E}_{\mathrm{tr}}$. Then, by reorganizing terms, we get

$$\bar{\Sigma}_e \beta^\star = \left(1 - \bar{\mu}_e^\top \beta^\star\right) \bar{\mu}_e.$$

Thus, using Equation equation 31 and equation 32, we have that

$$(\sigma_c^2 A A^\top + \sigma_e^2 B B^\top)\beta^\star = \left(1 - (A\mu_c + B\mu_e)^\top \beta^\star\right)(A\mu_c + B\mu_e).$$

Let $v := -\sigma^2 A A^\top \beta^\star + (1 - \mu_c^\top A \beta^\star) A \mu_c$. Then,

$$B\left(\sigma_e^2 I + \mu_e \mu_e^\top\right) B^\top \beta^\star - v = \left(1 - \mu_c^\top A^\top \beta^\star\right) B\mu_e + \mu_e^\top B^\top \beta^\star A \mu_c. \tag{33}$$

Similar to proof of Lemma C.3. in (Rosenfeld et al., 2021), we show that for all fixed $\beta^\star$ and $A$ Eq. equation 33 for all environments only holds (with probability 1) if $B = 0$. If $|\mathcal{E}_{\mathrm{tr}}| > d_e$. Then, by the degeneracy assumption of the training sets, there exists at least one environment for which Eq. equation 19 holds. Let $\tilde{\mu}$ and $\tilde{\sigma}^2$ be the mean of $Z_e$ and variance of $W_e$ for this environment. Then, we have that $\tilde{\mu} = \sum_{i=1}^{d_e} \alpha_i \mu_i$. By applying this linear combination to Eq. equation 33 for this environment, we get

$$B\left(\tilde{\sigma}^2 I + \tilde{\mu}\tilde{\mu}^\top\right) B^\top \beta^\star - v = \left(1 - \mu_c^\top A^\top \beta^\star\right) B \sum_{i=1}^{d_e} \alpha_i \mu_i + \left(\sum_{i=1}^{d_e} \alpha_i \mu_i\right)^\top B^\top \beta^\star A \mu_c$$

$$= \sum_{i=1}^{d_e} \alpha_i \left(\left(1 - \mu_c^\top A^\top \beta^\star\right) B \mu_i + \mu_i^\top B^\top \beta^\star A \mu_c\right)$$

$$= \sum_{i=1}^{d_e} \alpha_i \left(B\left(\sigma_i^2 I + \mu_i \mu_i^\top\right) B^\top \beta^\star - v\right), \tag{34}$$

where in the last identity, we applied Eq. equation 33 for all $i = 1, \ldots, d_e$. By rearranging the terms in Eq. equation 34, we get

$$B\left(\tilde{\sigma}^2 I + \tilde{\mu}\tilde{\mu}^\top - \sum_{i=1}^{d_e} \alpha_i \left(\sigma_i^2 I + \mu_i \mu_i^\top\right)\right) B^\top \beta^\star = \left(1 - \sum_{i=1}^{d_e} \alpha_i\right) v. \tag{35}$$

From the non-degeneracy condition equation 20, Eq. equation 35 is equivalent to

$$B\Gamma_\alpha B^\top \beta^\star = v, \tag{36}$$

where $\Gamma_\alpha$ is defined as

$$\Gamma_\alpha := \frac{\tilde{\sigma}^2 I + \tilde{\mu}\tilde{\mu}^\top - \sum_{i=1}^{d_e} \alpha_i \left(\sigma_i^2 I + \mu_i \mu_i^\top\right)}{1 - \sum_{i=1}^{d_e} \alpha_i}.$$

Note that $B$, $\beta^\star$, and $v$ are environment independent and $\Gamma_\alpha$ is an environment dependent matrix for which it holds that $\mathrm{rank}(\Gamma_\alpha) = d_e$ from the nondegeneracy condition equation 21. Thus, Eq. equation 36 holds if and only $v = B\Gamma_\alpha B^\top \beta^\star = 0$. Then, Eq. equation 33 reduces to

$$\left(1 - \mu_c^\top A^\top \beta^\star\right) B\mu_e + \beta^{\star\top} B\mu_e A\mu_c = 0$$

for all $e \in \mathcal{E}_{\mathrm{tr}}$. Thus, $B\mu_e = 0$ for all $e \in \mathcal{E}_{\mathrm{tr}}$, which holds if and only if $B = 0$ given that the span of $\mu_e$s is $\mathbb{R}^{d_e}$.

## D  THE ROLE OF THE EIGENSTRUCTURE OF $\mathcal{I}_e(\varphi)$

In this section, we elaborate on the discussions on the eigenstructure of $\mathcal{I}_e(\varphi)$, and in particular, its condition number in the examples of (Arjovsky et al., 2019) and (Rosenfeld et al., 2021).

### D.1  EXAMPLE 1 OF (ARJOVSKY ET AL., 2019)

Arjovsky et al. (2019) consider data that is generated according to the following SEM

$$X_1 \sim \mathcal{N}(0, \sigma^2),$$
$$Y = X_1 + Z_1,$$
$$X_2 = Y + Z_2,$$

where $Z_1 \sim \mathcal{N}(0, \sigma^2)$, $Z_2 \sim (0, 1)$, and $X_1$ are independent, and $X = \begin{bmatrix} X_1 \\ X_2 \end{bmatrix}$. Consider the following data representation

$$\varphi_c(X) = \begin{bmatrix} X_1 \\ cX_2 \end{bmatrix}. \tag{37}$$

Then,

$$\mathcal{I}(\varphi_c) = \mathbf{E}\left[\begin{bmatrix} X_1 \\ cX_2 \end{bmatrix}\begin{bmatrix} X_1 \\ cX_2 \end{bmatrix}^\top\right] = \begin{bmatrix} \sigma^2 & c\sigma^2 \\ c\sigma^2 & c^2(2\sigma^2 + 1) \end{bmatrix}.$$

We now find the eigenvalues of $\mathcal{I}(\varphi_c)$. That is, the solutions to $\det(\mathcal{I}(\varphi_c) - \lambda I) = 0$.

$$\lambda^2 - \lambda\left(\sigma^2 + c^2(2\sigma^2 + 1)\right) + c^2\sigma^2(2\sigma^2 + 1) - c^2\sigma^4 = 0.$$

Then,

$$\lambda = \frac{1}{2}\left(\sigma^2 + c^2(2\sigma^2 + 1) \pm \sqrt{\left(\sigma^2 + c^2(2\sigma^2 + 1)\right)^2 - 4c^2\sigma^2(\sigma^2 + 1)}\right)$$
$$= \frac{1}{2}\left(\sigma^2 + c^2(2\sigma^2 + 1) \pm \sqrt{\sigma^4 + c^4(2\sigma^2 + 1)^2 - 2c^2\sigma^2}\right).$$

Hence,

$$\kappa(\mathcal{I}(\varphi_c)) = \frac{\lambda_{\max}(\mathcal{I}(\varphi_c))}{\lambda_{\min}(\mathcal{I}(\varphi_c))}$$
$$= \frac{\sigma^2 + c^2(2\sigma^2 + 1) + \sqrt{\sigma^4 + c^4(2\sigma^2 + 1)^2 - 2c^2\sigma^2}}{\sigma^2 + c^2(2\sigma^2 + 1) - \sqrt{\sigma^4 + c^4(2\sigma^2 + 1)^2 - 2c^2\sigma^2}}$$
$$= \frac{\left(\sigma^2 + c^2(2\sigma^2 + 1) + \sqrt{\sigma^4 + c^4(2\sigma^2 + 1)^2 - 2c^2\sigma^2}\right)^2}{(\sigma^2 + c^2(2\sigma^2 + 1))^2 - (\sigma^4 + c^4(2\sigma^2 + 1)^2 - 2c^2\sigma^2)}$$
$$= \frac{1}{2(\sigma^2 + 1)}\left(\frac{1}{c}\sigma^2 + c(2\sigma^2 + 1) + \sqrt{\frac{1}{c^2}\sigma^4 + c^2(2\sigma^2 + 1)^2 - 2\sigma^2}\right)^2.$$

Finally,

$$\lim_{c \to \infty} \kappa(\mathcal{I}(\varphi_c)) = \lim_{c \to 0} \kappa(\mathcal{I}(\varphi_c)) = \infty.$$

It is worth noting that Arjovsky et al. (2019) discuss that $\|w - w_e^\star(\varphi)\|^2$ is poor discrepancy both for the invariant data representation, i.e., $c = 0$, and for a data representation that heavily rely on the spurious features, i.e., large $c$.

### D.2 EXAMPLE OF (ROSENFELD ET AL., 2021)

We show that in the setting that Rosenfeld et al. (2021) demonstrate that the invariance penalty is arbitrarily small, the risk is also inevitably arbitrarily small and is of the same order as the invariance penalty.

$$\varphi_\epsilon(X^e) = \begin{bmatrix} Z_c \\ 0 \end{bmatrix} + \begin{bmatrix} 0 \\ Z_e \end{bmatrix} \mathbb{1}_{\{Z_e \notin \mathcal{Z}_\epsilon\}}.$$

Here, $\mathcal{Z}_\epsilon$ is defined as

$$\mathcal{Z}_\epsilon := \bigcup_{e \in \mathcal{E}} \left( \mathcal{B}_r(\mu_e) \cup \mathcal{B}_r(-\mu_e) \right), \tag{38}$$

where $r := \sqrt{\epsilon \sigma_e^2 d_e}$ and $\mathcal{B}_r(\mu)$ denotes the $\ell_2$ ball of radius $r$ centered at $\mu$. Note that the invariant representation is given by

$$\varphi_{inv}(X^e) = \begin{bmatrix} Z_c \\ 0 \end{bmatrix}.$$

We compare the risk of the optimal predictor given the invariant and the Rosenfeld et al. (2021) representations.

#### D.2.1 INVARIANT REPRESENTATION

$$\mathcal{I}_e(\varphi_{inv}) = \mathbf{E}[\varphi_{inv}(X^e)\varphi_{inv}(X^e)^\top] = \mathbf{E}\begin{bmatrix} Z_c Z_c^\top & 0 \\ 0 & 0 \end{bmatrix} = \begin{bmatrix} \mu_c \mu_c^\top + \sigma_c^2 I & 0 \\ 0 & 0 \end{bmatrix}$$

Given that $\mathcal{I}_e(\varphi_{inv})$ is singular, we consider $\ell_2$ regularized LSE with regularization parameter $\gamma > 0$, i.e.,

$$w_{inv}^* = (\mathcal{I}_e(\varphi_{inv}) + \gamma I)^{-1} \mathbf{E}[\varphi_{inv}(X^e) Y_e].$$

Then,

$$w_{inv}^* = \begin{bmatrix} \mu_c \mu_c^\top + (\sigma_c^2 + \gamma) I & 0 \\ 0 & \gamma I \end{bmatrix}^{-1} \mathbf{E}\begin{bmatrix} Z_c Y \\ 0 \end{bmatrix} = \begin{bmatrix} (\mu_c \mu_c^\top + (\sigma_c^2 + \gamma) I)^{-1} \mu_c \\ 0 \end{bmatrix}$$

Recall from the proof of Lemma 1 that

$$R_e(w_{inv}^{*\top} \varphi_{inv}) = \mathbf{E}[\|Y\|^2] - \mathbf{E}[\varphi_{inv}(X) Y]^\top \mathcal{I}_e(\varphi_{inv}) \mathbf{E}[\varphi_{inv}(X) Y].$$

Then,

$$R_e(w_{inv}^{*\top} \varphi_{inv}) = 1 - \mu_c^\top (\mu_c \mu_c^\top + (\sigma_c^2 + \gamma) I)^{-1} \mu_c. \tag{39}$$

#### D.2.2 ROSENFELD ET AL. (2021) REPRESENTATION

We have that

$$\begin{aligned} \mathcal{I}_e(\varphi_\epsilon) &= \mathbf{E}[\varphi_\epsilon(X^e)\varphi_\epsilon(X^e)^\top] \\ &= \mathbf{E}\begin{bmatrix} Z_c Z_c^\top & 0 \\ 0 & 0 \end{bmatrix} + \mathbf{E}\left[ \begin{bmatrix} 0 & Z_c Z_e^\top \\ Z_e Z_c^\top & Z_e Z_e^\top \end{bmatrix} \mathbb{1}_{\{Z_e \notin \mathcal{Z}_\epsilon\}} \right]. \end{aligned}$$

Recall that the first term is equal to $\mathcal{I}_e(\varphi_{inv})$. For the second term, we have that

$$\mathbf{E}\left[\begin{bmatrix} 0 & Z_c Z_e^\top \\ Z_e Z_c^\top & Z_e Z_e^\top \end{bmatrix} \mathbb{1}_{\{Z_e \notin \mathcal{Z}_\epsilon\}}\right] = \mathbf{P}(Z_e \notin \mathcal{Z}_\epsilon) \mathbf{E}\left[\begin{bmatrix} 0 & Z_c Z_e^\top \\ Z_e Z_c^\top & Z_e Z_e^\top \end{bmatrix} \middle| Z_e \notin \mathcal{Z}_\epsilon\right].$$

Now define matrix $A$ and parameter $p$ as

$$A := \mathbf{E}\left[\begin{bmatrix} 0 & Z_c Z_e^\top \\ Z_e Z_c^\top & Z_e Z_e^\top \end{bmatrix} \middle| Z_e \notin \mathcal{Z}_\epsilon\right],$$

$$p := \mathbf{P}(Z_e \notin \mathcal{Z}_\epsilon).$$

Then,

$$\mathcal{I}_e(\varphi_\epsilon) = \mathcal{I}_e(\varphi_{inv}) + pA.$$

Also,

$$w_\epsilon^* = (\mathcal{I}_e(\varphi_\epsilon) + \gamma I)^{-1} \mathbf{E}[\varphi_\epsilon(X^e) Y_e]$$

$$= (\mathcal{I}_e(\varphi_{inv}) + \gamma I + pA)^{-1} \mathbf{E}\begin{bmatrix} Z_c Y \\ Z_e Y \mathbb{1}_{\{Z_e \notin \mathcal{Z}_\epsilon\}} \end{bmatrix}.$$

Let $a = \mathbf{E}[Z_e Y | Z_e \notin \mathcal{Z}_\epsilon]$. Then,

$$w_\epsilon^* = (\mathcal{I}_e(\varphi_{inv}) + \gamma I + pA)^{-1} \begin{bmatrix} \mu_c \\ pb \end{bmatrix}.$$

For an invertible matrix $S$, we have that

$$(S + \Delta)^{-1} = S^{-1} - S^{-1}\Delta S^{-1} + O(\|\Delta\|^2).$$

Thus,

$$w_\epsilon^* = ((\mathcal{I}_e(\varphi_{inv}) + \gamma I)^{-1} - p(\mathcal{I}_e(\varphi_{inv}) + \gamma I)^{-1} A (\mathcal{I}_e(\varphi_{inv}) + \gamma I)^{-1} + O(p^2)) \begin{bmatrix} \mu_c \\ pb \end{bmatrix}$$

$$= w_{inv}^* + p\left(\begin{bmatrix} 0 \\ b \end{bmatrix} - (\mathcal{I}_e(\varphi_{inv}) + \gamma I)^{-1} A w_{inv}^*\right) + O(p^2).$$

Then,

$$R_e(w_\epsilon^{*\top} \varphi_\epsilon) = 1 - \begin{bmatrix} \mu_c \\ pb \end{bmatrix}^\top w_\epsilon^*$$

$$= 1 - \mu_c^\top (\mu_c \mu_c^\top + (\sigma_c^2 + \gamma)I)^{-1} \mu_c + O(p^2).$$

That is,

$$R_e(w_\epsilon^{*\top} \varphi_\epsilon) - R_e(w_{inv}^{*\top} \varphi_{inv}) = O(p^2).$$

Recall that

$$\varphi_\epsilon(X^e) = \begin{bmatrix} Z_c \\ 0 \end{bmatrix} \mathbb{1}_{\{Z_e \in \mathcal{Z}_\epsilon\}} + \begin{bmatrix} Z_c \\ Z_e \end{bmatrix} \mathbb{1}_{\{Z_e \notin \mathcal{Z}_\epsilon\}}.$$

Here, $\mathcal{Z}_\epsilon$ is defined as

$$\mathcal{Z}_\epsilon := \bigcup_{e \in \mathcal{E}} (\mathcal{B}_r(\mu_e) \cup \mathcal{B}_r(-\mu_e)), \tag{40}$$

where $r := \sqrt{\epsilon \sigma_e^2 d_e}$ and $\mathcal{B}_r(\mu)$ denotes the $\ell_2$ ball of radius $r$ centered at $\mu$. Then,

$$\mathcal{I}_e(\varphi_\epsilon) = \mathbf{E}\left[\varphi_\epsilon(X^e) \varphi_\epsilon(X^e)^\top\right] = I_c + I_e.$$

where $I_c$ and $I_e$ are defined as

$$I_c := \begin{bmatrix} \mathbf{E}\left[Z_c Z_c^\top | Z_e \in \mathcal{Z}_\epsilon\right] & 0 \\ 0 & 0 \end{bmatrix} \mathbf{P}\left(Z_e \in \mathcal{Z}_\epsilon\right),$$

$$I_e := \mathbf{E}\left[\begin{bmatrix} Z_c \\ Z_e \end{bmatrix} \begin{bmatrix} Z_c \\ Z_e \end{bmatrix}^\top \middle| Z_e \in \mathcal{Z}_\epsilon\right] \mathbf{P}\left(Z_e \notin \mathcal{Z}_\epsilon\right).$$

Here, we establish a lower bound on the condition number of $\mathcal{I}_e(\varphi_\epsilon)$ in terms of the probability of event $\mathbb{1}_{\{Z_e \notin \mathcal{Z}_\epsilon\}}$. Using Weyl's inequality, we have that

$$\lambda_{\max}(\mathcal{I}_e(\varphi_\epsilon)) \geq \lambda_{\max}(I_c) + \lambda_{\min}(I_e),$$
$$\lambda_{\min}(\mathcal{I}_e(\varphi_\epsilon)) \leq \lambda_{\min}(I_c) + \lambda_{\max}(I_e).$$

As $I_e$ is positive semidefinite, $\lambda_{\min}(I_e) \geq 0$. Moreover, $\lambda_{\min}(I_c) = 0$. Then,

$$\lambda_{\max}(\mathcal{I}_e(\varphi_\epsilon)) \geq \lambda_{\max}(I_c),$$
$$\lambda_{\min}(\mathcal{I}_e(\varphi_\epsilon)) \leq \lambda_{\max}(I_e).$$

For the first term, we have

$$
\begin{aligned}
\lambda_{\max}(I_c) &\geq \frac{1}{d_e + d_c} \mathrm{tr}(I_c) \\
&= \frac{1}{d_e + d_c} \mathbf{E}\left[\|Z_c\|^2 | Z_e \in \mathcal{Z}_\epsilon\right] \mathbf{P}\left(Z_e \in \mathcal{Z}_\epsilon\right) \\
&= \frac{1}{d_e + d_c} \mathbf{E}\left[\|\mu_c\|^2 Y^2 + \|W_c\|^2 + 2\mu_c^\top W_c Y | Z_e \in \mathcal{Z}_\epsilon\right] \mathbf{P}\left(Z_e \in \mathcal{Z}_\epsilon\right) \\
&= \frac{1}{d_e + d_c} (\|\mu_c\|^2 + d_c \sigma_c^2) \mathbf{P}\left(Z_e \in \mathcal{Z}_\epsilon\right), \tag{41}
\end{aligned}
$$

where the last identity follows from the fact that $Y^2 = 1$ almost surely, and the fact that $W_c$ is independent of $Y$ and $W_e$, and hence is independent of the event $\mathbb{1}_{\{Z_e \notin \mathcal{Z}_\epsilon\}}$.

For the second term, we have

$$
\begin{aligned}
\lambda_{\max}(I_e) &\leq \mathrm{tr}(I_c) \\
&= \mathbf{E}\left[\|Z_c\|^2 + \|Z_e\|^2 | Z_e \notin \mathcal{Z}_\epsilon\right] \mathbf{P}\left(Z_e \notin \mathcal{Z}_\epsilon\right) \\
&= \left(\|\mu_c\|^2 + d_c \sigma_c^2 + \mathbf{E}\left[\|Z_e\|^2 | Z_e \notin \mathcal{Z}_\epsilon\right]\right) \mathbf{P}\left(Z_e \notin \mathcal{Z}_\epsilon\right),
\end{aligned}
$$

where the last identity follows similarly as Equation equation 41. Then,

$$
\begin{aligned}
\mathbf{E}\left[\|Z_e\|^2 | Z_e \notin \mathcal{Z}_\epsilon\right] &= \mathbf{E}\left[\|\mu_e\|^2 Y^2 + \|W_e\|^2 + 2\mu_e^\top W_e Y | Z_e \notin \mathcal{Z}_\epsilon\right] \\
&= \|\mu_e\|^2 + \mathbf{E}\left[\|W_e\|^2 + 2\mu_e^\top W_e Y | Z_e \notin \mathcal{Z}_\epsilon\right].
\end{aligned}
$$

We have that

$$\mathbf{E}\left[\|W_e\|^2 | Z_e \notin \mathcal{Z}_\epsilon\right] \leq d_e \sigma_e^2.$$

Moreover, using Cauchy-Schwarz inequality, we get

$$\mathbf{E}\left[\mu_e^\top W_e Y | Z_e \notin \mathcal{Z}_\epsilon\right] \leq \|\mu_e\| \mathbf{E}\left[\|W_e\| |Y| | Z_e \notin \mathcal{Z}_\epsilon\right] \leq \|\mu_e\| \sqrt{d_e \sigma_e^2}.$$

Hence,

$$\lambda_{\max}(I_e) \leq \left(\|\mu_c\|^2 + d_c \sigma_c^2 + (\|\mu_e\| + \sqrt{d_e \sigma_e^2})^2\right) \mathbf{P}\left(Z_e \notin \mathcal{Z}_\epsilon\right).$$

Thus,

$$
\begin{aligned}
\kappa(\mathcal{I}_e(\varphi_\epsilon)) &\geq \frac{\lambda_{\max}(I_c)}{\lambda_{\max}(I_e)} \\
&\geq \frac{(\|\mu_c\|^2 + d_c \sigma_c^2) \mathbf{P}\left(Z_e \in \mathcal{Z}_\epsilon\right) / (d_e + d_c)}{\left(\|\mu_c\|^2 + d_c \sigma_c^2 + (\|\mu_e\| + \sqrt{d_e \sigma_e^2})^2\right) \mathbf{P}\left(Z_e \notin \mathcal{Z}_\epsilon\right)} \\
&= \frac{\|\mu_c\|^2 + d_c \sigma_c^2}{(d_e + d_c)\left(\|\mu_c\|^2 + d_c \sigma_c^2 + (\|\mu_e\| + \sqrt{d_e \sigma_e^2})^2\right)} \left(\frac{1}{\mathbf{P}\left(Z_e \notin \mathcal{Z}_\epsilon\right)} - 1\right).
\end{aligned}
$$

Note that (Rosenfeld et al., 2021, Lemma F.3.) show that

$$\mathbf{P}\left(Z_e \notin \mathcal{Z}_\epsilon\right) \leq p_{\epsilon, e},$$

where

$$p_{\epsilon,e} := \exp\left(-\frac{1}{8}\min\{\epsilon - 1, (\epsilon - 1)^2\}\right) \tag{42}$$

Then,

$$\kappa(\mathcal{I}_e(\varphi_\epsilon)) \geq \frac{\|\mu_c\|^2 + d_c\sigma_c^2}{(d_e + d_c)\left(\|\mu_c\|^2 + d_c\sigma_c^2 + (\|\mu_e\| + \sqrt{d_e\sigma_e^2})^2\right)}\left(\frac{1}{p_{\epsilon,e}} - 1\right).$$

Rosenfeld et al. (2021) show that the invariance penalty of (Arjovsky et al., 2019) is no greater than $O(p_{\epsilon,e}^2)$, which can be made arbitrarily small by choosing appropriately large $\epsilon$. However, for such choices of $\epsilon$, matrix $\mathcal{I}_e(\varphi_\epsilon)$ is ill-conditioned. In particular,

$$\lim_{\epsilon\to\infty}\kappa(\mathcal{I}_e(\varphi_\epsilon)) = \infty.$$

### D.3 EXTENDED COLORED MNIST EXPERIMENTS

The model and all its respective parameters are replicated from Arjovsky et al. (2019)'s GitHub repository[3]. The penalty coefficient for IRMv2 is set at $\lambda = 10$ and for $\lambda_0 = 10^{-5}$ for IRMv1A.

Figure 2 depicts demonstrate the significance of the dimension of the representation in relation to $\mathcal{I}(\varphi)$ being ill-conditioned, and, in turn, the test accuracy of IRMv1, v1A, and v2. In particular, by allowing the representation to be overly flexible

## E FULL LINEARUNITTESTS EXPERIMENTS

In this section, we provide more detailed tables for the numerical experiments on LinearUnitTests data sets. In addition to ERM, IRMv1, IRMv1A, and IRMv2, we include two other invariance discovery methods, namely Inter-environmental Gradient Alignment (IGA) (Koyama & Yamaguchi, 2020) and AND-Mask (Parascandolo et al., 2020), considered in (Aubin et al., 2020). The IGA method seeks to elicit invariant predictors by an invariance penalty in terms of the variance of the risk under different environments. The AND-Mask method, at each step of the training process, updates the model using the direction where gradient (of the loss) signs agree across environments.

All experiments are run using default parameters of the experiments of Aubin et al. (2020) available on their GitHub repository.[4]

**We note that in the majority of these experiment (13 out of 18) one of our proposed methods IRMv1A or IRMv2 outperforms all other methods.**

## F DETAILS OF THE HEALTHYGUTTESTS EXPERIMENTS

We summarize the details of each data set of the HealthyGutTests in Table 7.

|  | Sample Size | Geographical Region | Health Status Studied |
|---|---|---|---|
| Feng (2015) | 145 | Austria | Colorectal Cancer, Advanced Adenoma, Obesity, Overweight, Healthy |
| He (2017) | 98 | China | Crohns Disease, Obesity, Overweight, Underweight, Healthy |
| Jie (2017) | 281 | China | Atherosclerotic Cardiovascular Disease, Obesity, Overweight, Underweight, Healthy |
| Karlsson (2013) | 133 | Europe | Type 2 Diabetes, Impaired Glucose Tolerance, Obesity, Overweight, Underweight, Healthy |
| Le Chatelier (2013) | 112 | Denmark | Obesity, Overweight, Underweight, Healthy |
| Nielsen (2014) | 226 | Europe | Crohns Disease, Ulcerative Colitis, Obesity, Overweight, Underweight, Healthy |
| Qin (2012) | 297 | China | Type 2 Diabetes, Obesity, Overweight, Underweight, Healthy |
| Vogtmann (2016) | 99 | USA | Colorectal Cancer, Obesity, Overweight, Underweight, Healthy |
| Zeller (2014) | 196 | Europe | Colorectal Cancer, Obesity, Overweight, Underweight, Healthy |
| Zhang (2015) | 183 | China | Rheumatoid Arthritis, Obesity, Overweight, Underweight, Healthy |

Table 7: Details of the *HealthyGutTests* data sets.

---

[3] https://github.com/facebookresearch/InvariantRiskMinimization
[4] https://github.com/facebookresearch/InvarianceUnitTests

| | ANDMask | ERM | IGA | IRMv1A | IRMv1 | IRMv2 | Oracle |
|---|---|---|---|---|---|---|---|
| Example1.E0 | $0.14 \pm 0.03$ | $2.01 \pm 0.27$ | $4.68 \pm 0.72$ | $0.07 \pm 0.01$ | $0.13 \pm 0.01$ | $\mathbf{0.05 \pm 0.00}$ | $0.05 \pm 0.00$ |
| Example1.E1 | $\mathbf{11.56 \pm 0.20}$ | $15.37 \pm 0.52$ | $19.61 \pm 1.36$ | $13.81 \pm 1.11$ | $11.60 \pm 0.19$ | $12.15 \pm 0.81$ | $11.25 \pm 0.02$ |
| Example1.E2 | $\mathbf{20.38 \pm 0.27}$ | $25.82 \pm 0.89$ | $31.13 \pm 2.30$ | $24.74 \pm 2.20$ | $20.43 \pm 0.17$ | $21.56 \pm 1.11$ | $19.66 \pm 0.38$ |
| Example1s.E0 | $0.12 \pm 0.05$ | $1.96 \pm 0.29$ | $3.88 \pm 0.17$ | $0.07 \pm 0.01$ | $0.13 \pm 0.01$ | $\mathbf{0.06 \pm 0.01}$ | $0.05 \pm 0.00$ |
| Example1s.E1 | $13.68 \pm 1.00$ | $15.47 \pm 0.87$ | $18.21 \pm 0.55$ | $14.35 \pm 1.62$ | $\mathbf{11.72 \pm 0.08}$ | $13.30 \pm 1.87$ | $11.03 \pm 0.27$ |
| Example1s.E2 | $24.13 \pm 1.40$ | $25.69 \pm 1.06$ | $28.92 \pm 0.53$ | $25.32 \pm 2.61$ | $\mathbf{20.44 \pm 0.32}$ | $23.38 \pm 3.09$ | $20.53 \pm 0.28$ |
| Example2.E0 | $0.40 \pm 0.03$ | $0.40 \pm 0.01$ | $0.43 \pm 0.00$ | $\mathbf{0.30 \pm 0.09}$ | $0.43 \pm 0.00$ | $0.43 \pm 0.09$ | $0.00 \pm 0.00$ |
| Example2.E1 | $0.47 \pm 0.04$ | $0.47 \pm 0.01$ | $0.50 \pm 0.00$ | $\mathbf{0.34 \pm 0.11}$ | $0.50 \pm 0.00$ | $0.35 \pm 0.05$ | $0.00 \pm 0.00$ |
| Example2.E2 | $0.40 \pm 0.03$ | $0.39 \pm 0.00$ | $0.42 \pm 0.00$ | $0.31 \pm 0.09$ | $0.42 \pm 0.00$ | $\mathbf{0.23 \pm 0.02}$ | $0.00 \pm 0.00$ |
| Example2s.E0 | $0.43 \pm 0.00$ | $0.43 \pm 0.00$ | $0.43 \pm 0.00$ | $\mathbf{0.25 \pm 0.17}$ | $0.43 \pm 0.00$ | $0.45 \pm 0.06$ | $0.00 \pm 0.00$ |
| Example2s.E1 | $0.50 \pm 0.00$ | $0.50 \pm 0.00$ | $0.50 \pm 0.00$ | $\mathbf{0.29 \pm 0.20}$ | $0.50 \pm 0.00$ | $0.40 \pm 0.05$ | $0.00 \pm 0.00$ |
| Example2s.E2 | $0.42 \pm 0.00$ | $0.42 \pm 0.00$ | $0.42 \pm 0.00$ | $\mathbf{0.26 \pm 0.17}$ | $0.42 \pm 0.00$ | $0.29 \pm 0.07$ | $0.00 \pm 0.00$ |
| Example3.E0 | $0.34 \pm 0.23$ | $0.35 \pm 0.21$ | $0.34 \pm 0.22$ | $\mathbf{0.25 \pm 0.02}$ | $0.36 \pm 0.20$ | $0.41 \pm 0.03$ | $0.01 \pm 0.00$ |
| Example3.E1 | $0.36 \pm 0.21$ | $0.40 \pm 0.14$ | $0.38 \pm 0.17$ | $\mathbf{0.26 \pm 0.02}$ | $0.37 \pm 0.18$ | $0.39 \pm 0.00$ | $0.01 \pm 0.00$ |
| Example3.E2 | $0.34 \pm 0.23$ | $0.36 \pm 0.20$ | $0.35 \pm 0.21$ | $\mathbf{0.26 \pm 0.02}$ | $0.36 \pm 0.20$ | $0.39 \pm 0.00$ | $0.01 \pm 0.00$ |
| Example3s.E0 | $\mathbf{0.28 \pm 0.17}$ | $0.35 \pm 0.21$ | $0.35 \pm 0.21$ | $0.34 \pm 0.11$ | $0.36 \pm 0.20$ | $0.41 \pm 0.01$ | $0.01 \pm 0.00$ |
| Example3s.E1 | $0.47 \pm 0.06$ | $0.41 \pm 0.14$ | $0.40 \pm 0.15$ | $\mathbf{0.34 \pm 0.11}$ | $0.38 \pm 0.19$ | $0.41 \pm 0.01$ | $0.01 \pm 0.00$ |
| Example3s.E2 | $0.38 \pm 0.17$ | $0.36 \pm 0.20$ | $0.36 \pm 0.20$ | $\mathbf{0.34 \pm 0.11}$ | $0.36 \pm 0.20$ | $0.41 \pm 0.01$ | $0.01 \pm 0.00$ |

Table 3: Test errors for all algorithms and examples with $(d_{\text{inv}}, d_{\text{spu}}, d_{\text{env}}) = (5, 5, 3)$. The errors for Examples 1 and 1s are in MSE and all others are classification error. The empirical mean and the standard deviation are computed using 10 independent experiments. An 's' indicates the scrambled variation of its corresponding problem setting.

| | ANDMask | ERM | IGA | IRMv1A | IRMv1 | IRMv2 | Oracle |
|---|---|---|---|---|---|---|---|
| Example1 | $\mathbf{10.69 \pm 0.17}$ | $14.40 \pm 0.56$ | $18.48 \pm 1.46$ | $12.87 \pm 1.11$ | $10.72 \pm 0.13$ | $11.26 \pm 0.64$ | $10.32 \pm 0.13$ |
| Example1s | $12.64 \pm 0.82$ | $14.37 \pm 0.74$ | $17.00 \pm 0.42$ | $13.24 \pm 1.41$ | $\mathbf{10.76 \pm 0.14}$ | $12.25 \pm 1.66$ | $10.53 \pm 0.18$ |
| Example2 | $0.42 \pm 0.03$ | $0.42 \pm 0.01$ | $0.45 \pm 0.00$ | $\mathbf{0.31 \pm 0.10}$ | $0.45 \pm 0.00$ | $0.34 \pm 0.05$ | $0.00 \pm 0.00$ |
| Example2s | $0.45 \pm 0.00$ | $0.45 \pm 0.00$ | $0.45 \pm 0.00$ | $\mathbf{0.27 \pm 0.18}$ | $0.45 \pm 0.00$ | $0.38 \pm 0.06$ | $0.00 \pm 0.00$ |
| Example3 | $0.34 \pm 0.22$ | $0.37 \pm 0.18$ | $0.36 \pm 0.20$ | $\mathbf{0.26 \pm 0.02}$ | $0.36 \pm 0.19$ | $0.40 \pm 0.01$ | $0.01 \pm 0.00$ |
| Example3s | $0.38 \pm 0.13$ | $0.37 \pm 0.18$ | $0.37 \pm 0.19$ | $\mathbf{0.34 \pm 0.11}$ | $0.37 \pm 0.19$ | $0.41 \pm 0.01$ | $0.01 \pm 0.00$ |

Table 4: Test errors averaged for each experiment.

| | ANDMask | ERM | IGA | IRMv1A | IRMv1 | IRMv2 | Oracle |
|---|---|---|---|---|---|---|---|
| Example1.E0 | $0.13 \pm 0.03$ | $1.98 \pm 0.29$ | $4.67 \pm 0.74$ | $0.07 \pm 0.01$ | $0.12 \pm 0.01$ | $0.05 \pm 0.00$ | $0.05 \pm 0.00$ |
| Example1.E1 | $10.38 \pm 0.43$ | $7.80 \pm 0.51$ | $10.05 \pm 0.75$ | $13.82 \pm 1.26$ | $10.16 \pm 0.10$ | $12.01 \pm 0.68$ | $11.23 \pm 0.20$ |
| Example1.E2 | $18.45 \pm 0.63$ | $12.27 \pm 1.02$ | $14.13 \pm 1.39$ | $24.59 \pm 2.26$ | $18.12 \pm 0.24$ | $21.36 \pm 0.92$ | $20.19 \pm 0.13$ |
| Example1s.E0 | $0.12 \pm 0.05$ | $1.91 \pm 0.27$ | $3.82 \pm 0.12$ | $0.07 \pm 0.01$ | $0.12 \pm 0.01$ | $0.06 \pm 0.01$ | $0.05 \pm 0.00$ |
| Example1s.E1 | $12.14 \pm 0.26$ | $7.83 \pm 0.55$ | $9.26 \pm 0.52$ | $13.85 \pm 1.37$ | $10.15 \pm 0.10$ | $12.87 \pm 1.81$ | $11.27 \pm 0.02$ |
| Example1s.E2 | $21.33 \pm 0.27$ | $12.22 \pm 1.17$ | $13.23 \pm 0.91$ | $24.37 \pm 2.43$ | $17.80 \pm 0.24$ | $22.73 \pm 3.27$ | $19.96 \pm 0.17$ |
| Example2.E0 | $0.05 \pm 0.01$ | $0.04 \pm 0.00$ | $0.05 \pm 0.00$ | $0.12 \pm 0.06$ | $0.05 \pm 0.00$ | $0.36 \pm 0.11$ | $0.00 \pm 0.00$ |
| Example2.E1 | $0.03 \pm 0.00$ | $0.03 \pm 0.00$ | $0.03 \pm 0.00$ | $0.11 \pm 0.06$ | $0.03 \pm 0.00$ | $0.26 \pm 0.09$ | $0.00 \pm 0.00$ |
| Example2.E2 | $0.01 \pm 0.00$ | $0.01 \pm 0.00$ | $0.01 \pm 0.00$ | $0.10 \pm 0.07$ | $0.01 \pm 0.00$ | $0.15 \pm 0.05$ | $0.00 \pm 0.00$ |
| Example2s.E0 | $0.05 \pm 0.00$ | $0.05 \pm 0.00$ | $0.05 \pm 0.00$ | $0.05 \pm 0.03$ | $0.05 \pm 0.00$ | $0.31 \pm 0.19$ | $0.00 \pm 0.00$ |
| Example2s.E1 | $0.03 \pm 0.00$ | $0.03 \pm 0.00$ | $0.03 \pm 0.00$ | $0.04 \pm 0.03$ | $0.03 \pm 0.00$ | $0.22 \pm 0.14$ | $0.00 \pm 0.00$ |
| Example2s.E2 | $0.01 \pm 0.00$ | $0.01 \pm 0.00$ | $0.01 \pm 0.00$ | $0.03 \pm 0.02$ | $0.01 \pm 0.00$ | $0.13 \pm 0.09$ | $0.00 \pm 0.00$ |
| Example3.E0 | $0.00 \pm 0.01$ | $0.00 \pm 0.00$ | $0.00 \pm 0.00$ | $0.49 \pm 0.00$ | $0.00 \pm 0.00$ | $0.35 \pm 0.10$ | $0.01 \pm 0.00$ |
| Example3.E1 | $0.00 \pm 0.00$ | $0.00 \pm 0.00$ | $0.00 \pm 0.00$ | $0.49 \pm 0.00$ | $0.00 \pm 0.00$ | $0.29 \pm 0.01$ | $0.01 \pm 0.00$ |
| Example3.E2 | $0.01 \pm 0.01$ | $0.00 \pm 0.00$ | $0.00 \pm 0.00$ | $0.49 \pm 0.00$ | $0.00 \pm 0.00$ | $0.29 \pm 0.01$ | $0.01 \pm 0.00$ |
| Example3s.E0 | $0.18 \pm 0.09$ | $0.00 \pm 0.00$ | $0.00 \pm 0.00$ | $0.49 \pm 0.00$ | $0.00 \pm 0.00$ | $0.30 \pm 0.03$ | $0.01 \pm 0.00$ |
| Example3s.E1 | $0.00 \pm 0.00$ | $0.00 \pm 0.00$ | $0.00 \pm 0.00$ | $0.49 \pm 0.00$ | $0.00 \pm 0.00$ | $0.29 \pm 0.03$ | $0.01 \pm 0.00$ |
| Example3s.E2 | $0.02 \pm 0.02$ | $0.00 \pm 0.00$ | $0.00 \pm 0.00$ | $0.49 \pm 0.00$ | $0.00 \pm 0.00$ | $0.30 \pm 0.03$ | $0.01 \pm 0.00$ |

Table 5: Training errors for all algorithms and examples with $(d_{\text{inv}}, d_{\text{spu}}, d_{\text{env}}) = (5, 5, 3)$. The errors for Examples 1 and 1s are in MSE and all others are classification error. The empirical mean and the standard deviation are computed using 10 independent experiments. An 's' indicates the scrambled variation of its corresponding problem setting.

|  | ANDMask | ERM | IGA | IRMv1A | IRMv1 | IRMv2 | Oracle |
|---|---|---|---|---|---|---|---|
| Example1 | $9.65 \pm 0.36$ | $7.35 \pm 0.61$ | $9.62 \pm 0.96$ | $12.82 \pm 1.17$ | $9.47 \pm 0.12$ | $11.14 \pm 0.53$ | $10.49 \pm 0.11$ |
| Example1s | $11.20 \pm 0.19$ | $7.32 \pm 0.66$ | $8.77 \pm 0.52$ | $12.76 \pm 1.27$ | $9.36 \pm 0.12$ | $11.89 \pm 1.70$ | $10.43 \pm 0.06$ |
| Example2 | $0.03 \pm 0.00$ | $0.03 \pm 0.00$ | $0.03 \pm 0.00$ | $0.11 \pm 0.06$ | $0.03 \pm 0.00$ | $0.26 \pm 0.08$ | $0.00 \pm 0.00$ |
| Example2s | $0.03 \pm 0.00$ | $0.03 \pm 0.00$ | $0.03 \pm 0.00$ | $0.04 \pm 0.03$ | $0.03 \pm 0.00$ | $0.22 \pm 0.14$ | $0.00 \pm 0.00$ |
| Example3 | $0.00 \pm 0.00$ | $0.00 \pm 0.00$ | $0.00 \pm 0.00$ | $0.49 \pm 0.00$ | $0.00 \pm 0.00$ | $0.31 \pm 0.04$ | $0.01 \pm 0.00$ |
| Example3s | $0.07 \pm 0.04$ | $0.00 \pm 0.00$ | $0.00 \pm 0.00$ | $0.49 \pm 0.00$ | $0.00 \pm 0.00$ | $0.30 \pm 0.03$ | $0.01 \pm 0.00$ |

Table 6: Training errors averaged for each experiment.

### F.1 EXPERIMENT DETAILS

#### F.1.1 NEURAL NETWORK ARCHITECTURE AND HYPERPARAMETER TUNING

The neural network considered consists of 5 layers with input layer $d_X \times d_h$, hidden layers of size $d_h \times d_h/2$, $d_h/2 \times d_h/2$, $d_h/2 \times d_h/4$, respectively, and the output layer $d_h/4 \times d_\varphi$. We use the ReLU action function and the dropout parameter of each layer is set as $p$. We treat $d_h, d_\varphi, p$, the learning rate, and the invariance penalty coefficient as hyperparameters, which we optimize using the validation accuracy. More specifically, we randomly generate 200 hyperparameters. For each set of parameters, we independently repeat 5 times training the model over 500 epochs. We then select the model with best average validation accuracy (over 5 experiments), and report the average test accuracy in Table 2.

For each data set in Table 2 that is considered as the test set, we treat each of the remaining data sets as an environment. Before doing so, we split each of training data sets into $(0.8, 0.2)$ split of training and validation sets, respectively. We then concatenate all the validation subsets into a unified validation set for model selection.

## G   FULL DOMAINBED EXPERIMENTS

### G.1 EXPERIMENT SUMMARY

DomainBed is an extensive framework released by Gulrajani & Lopez-Paz (2020) to test domain generalization algorithms for image classification tasks on various benchmark data sets. In a series of experiments, Gulrajani & Lopez-Paz (2020) show that enabled by data augmentation various state-of-the-art generalization methods perform similar to each other and ERM on several benchmark data sets.

Although the integration of additional data sets and algorithms to DomainBed is straightforward, we note that performing an extensive set of experiments requires significant computational resources as also pointed out by Krueger et al. (2020). For this reason, we limit the scope of our experiments to the comparison of ERM, IRMv1, IRMv1A, and IRMv2.

Similar to Gulrajani & Lopez-Paz (2020), we observe that no method significantly outperforms others on any of the benchmark data sets (see Table 8). For a complete set of results on DomainBed with various model selection methods, we refer the reader to Appendix G. As these data sets are image based and equipped with data augmentation, they may not provide comprehensive insight on the strengths and weaknesses of domain generalization algorithms on other modes of data, e.g., gathered in real-world applications.

| Algorithm | ColoredMNIST | RotatedMNIST | PACS | VLCS | Avg |
|-----------|--------------|--------------|------|------|-----|
| ERM | $51.7 \pm 0.1$ | $96.7 \pm 0.0$ | $81.1 \pm 0.1$ | $\mathbf{78.8 \pm 0.4}$ | $\mathbf{77.0}$ |
| IRMv1 | $\mathbf{51.8 \pm 0.2}$ | $95.2 \pm 0.4$ | $78.6 \pm 1.0$ | $76.0 \pm 0.5$ | $75.4$ |
| IRMv1A | $50.9 \pm 0.1$ | $64.7 \pm 20.1$ | $80.9 \pm 0.0$ | $77.3 \pm 0.2$ | $68.4$ |
| IRMv2 | $50.8 \pm 0.4$ | $\mathbf{97.1 \pm 0.0}$ | $\mathbf{82.6 \pm 0.9}$ | $76.5 \pm 0.4$ | $76.8$ |

Table 8: The test accuracy of ERM and different implementations of IRM on benchmark data-sets. Model selection of the DomainBed is chosen as training-domain validation set.

| Algorithm | +90% | +80% | -90% | Avg |
|-----------|------|------|------|-----|
| ERM | $\mathbf{72.8 \pm 0.1}$ | $72.6 \pm 0.2$ | $9.8 \pm 0.0$ | $51.7$ |
| IRMv1 | $72.5 \pm 0.3$ | $72.9 \pm 0.1$ | $\mathbf{9.9 \pm 0.1}$ | $\mathbf{51.8}$ |
| IRMv1A | $70.7 \pm 0.3$ | $72.3 \pm 0.5$ | $9.7 \pm 0.0$ | $50.9$ |
| IRMv2 | $69.8 \pm 0.8$ | $\mathbf{72.9 \pm 0.3}$ | $9.8 \pm 0.1$ | $50.8$ |

Table 9: Benchmark: ColoredMNIST, Model selection: training-domain validation set.

## G.2 FULL DOMAINBED RESULTS

| Algorithm | A | C | P | S | Avg |
|-----------|---|---|---|---|-----|
| ERM | $84.5 \pm 1.6$ | $77.1 \pm 0.8$ | $\mathbf{96.9 \pm 0.3}$ | $65.8 \pm 1.9$ | $81.1$ |
| IRMv1 | $77.0 \pm 3.0$ | $76.7 \pm 1.1$ | $96.4 \pm 0.4$ | $64.4 \pm 0.3$ | $78.6$ |
| IRMv1A | $82.6 \pm 0.5$ | $\mathbf{77.7 \pm 0.7}$ | $96.6 \pm 0.4$ | $66.7 \pm 0.5$ | $80.9$ |
| IRMv2 | $\mathbf{86.0 \pm 0.9}$ | $76.6 \pm 0.7$ | $\mathbf{96.9 \pm 0.0}$ | $\mathbf{70.8 \pm 2.0}$ | $\mathbf{82.6}$ |

Table 11: Benchmark: PACS, Model selection: training-domain validation set.

| Algorithm | 0 | 15 | 30 | 45 | 60 | 75 | Avg |
|-----------|---|----|----|----|----|----|-----|
| ERM | $90.3 \pm 1.8$ | $97.8 \pm 0.0$ | $98.2 \pm 0.1$ | $\mathbf{98.2 \pm 0.1}$ | $97.8 \pm 0.2$ | $93.5 \pm 0.4$ | $96.0$ |
| IRMv1 | $89.6 \pm 2.1$ | $96.0 \pm 0.5$ | $97.9 \pm 0.1$ | $97.2 \pm 0.0$ | $97.0 \pm 0.2$ | $90.9 \pm 0.5$ | $94.8$ |
| IRMv1A | $75.9 \pm 5.9$ | $64.2 \pm 22.4$ | $59.4 \pm 26.1$ | $59.8 \pm 26.3$ | $59.0 \pm 26.7$ | $55.9 \pm 22.5$ | $62.4$ |
| IRMv2 | $\mathbf{94.1 \pm 0.0}$ | $\mathbf{98.1 \pm 0.3}$ | $\mathbf{98.5 \pm 0.1}$ | $\mathbf{98.2 \pm 0.1}$ | $\mathbf{98.3 \pm 0.0}$ | $\mathbf{94.4 \pm 0.3}$ | $\mathbf{97.0}$ |

Table 15: Benchmark: RotatedMNIST, Model selection: leave-one-domain-out cross-validation.

| Algorithm | A | C | P | S | Avg |
|-----------|---|---|---|---|-----|
| ERM | $79.7 \pm 0.6$ | $73.0 \pm 3.8$ | $\mathbf{97.1 \pm 0.9}$ | $62.1 \pm 1.5$ | $78.0$ |
| IRMv1 | $67.4 \pm 6.7$ | $72.3 \pm 4.3$ | $87.7 \pm 5.7$ | $64.1 \pm 4.5$ | $72.9$ |
| IRMv1A | $78.8 \pm 3.2$ | $\mathbf{78.9 \pm 0.9}$ | $96.0 \pm 0.2$ | $42.3 \pm 16.3$ | $74.0$ |
| IRMv2 | $\mathbf{86.3 \pm 0.3}$ | $76.8 \pm 0.5$ | $97.0 \pm 0.4$ | $\mathbf{69.7 \pm 2.6}$ | $\mathbf{82.5}$ |

Table 16: Benchmark: PACS, Model selection: leave-one-domain-out cross-validation.

| Algorithm | 0 | 15 | 30 | 45 | 60 | 75 | Avg |
|---|---|---|---|---|---|---|---|
| ERM | $93.1 \pm 0.1$ | $97.8 \pm 0.0$ | $98.4 \pm 0.0$ | $98.3 \pm 0.1$ | $98.2 \pm 0.0$ | $94.3 \pm 0.1$ | 96.7 |
| IRMv1 | $89.6 \pm 2.1$ | $96.8 \pm 0.1$ | $97.9 \pm 0.1$ | $97.8 \pm 0.1$ | $97.5 \pm 0.1$ | $91.6 \pm 0.0$ | 95.2 |
| IRMv1A | $75.9 \pm 5.9$ | $71.1 \pm 17.7$ | $60.8 \pm 25.1$ | $60.4 \pm 25.8$ | $60.2 \pm 25.8$ | $59.8 \pm 20.5$ | 64.7 |
| IRMv2 | $\mathbf{94.1 \pm 0.0}$ | $\mathbf{98.2 \pm 0.0}$ | $\mathbf{98.5 \pm 0.1}$ | $\mathbf{98.4 \pm 0.1}$ | $\mathbf{98.3 \pm 0.0}$ | $\mathbf{95.1 \pm 0.2}$ | $\mathbf{97.1}$ |

Table 10: Benchmark: RotatedMNIST, Model selection: training-domain validation set.

| Algorithm | C | L | S | V | Avg |
|---|---|---|---|---|---|
| ERM | $\mathbf{97.4 \pm 0.1}$ | $65.0 \pm 0.9$ | $\mathbf{74.3 \pm 1.1}$ | $\mathbf{78.7 \pm 0.1}$ | $\mathbf{78.8}$ |
| IRM | $96.3 \pm 0.6$ | $61.7 \pm 0.3$ | $70.1 \pm 0.1$ | $76.0 \pm 1.8$ | 76.0 |
| IRMA | $96.9 \pm 0.8$ | $64.8 \pm 0.0$ | $70.7 \pm 1.4$ | $77.0 \pm 0.4$ | 77.3 |
| IRMv2 | $96.6 \pm 1.1$ | $\mathbf{65.4 \pm 1.5}$ | $73.5 \pm 0.5$ | $70.6 \pm 2.4$ | 76.5 |

Table 12: Benchmark: VLCS, Model selection: training-domain validation set.

| Algorithm | 0 | 15 | 30 | 45 | 60 | 75 | Avg |
|---|---|---|---|---|---|---|---|
| ERM | $92.5 \pm 0.6$ | $97.8 \pm 0.0$ | $97.9 \pm 0.1$ | $97.9 \pm 0.2$ | $\mathbf{98.3 \pm 0.1}$ | $94.3 \pm 0.1$ | 96.4 |
| IRMv1 | $89.6 \pm 2.1$ | $96.8 \pm 0.1$ | $98.0 \pm 0.2$ | $97.5 \pm 0.3$ | $97.5 \pm 0.1$ | $91.6 \pm 0.0$ | 95.2 |
| IRMv1A | $77.9 \pm 7.4$ | $71.1 \pm 17.7$ | $59.4 \pm 26.1$ | $59.8 \pm 26.3$ | $59.0 \pm 26.7$ | $55.9 \pm 22.5$ | 63.9 |
| IRMv2 | $\mathbf{94.7 \pm 0.4}$ | $\mathbf{98.0 \pm 0.2}$ | $\mathbf{98.5 \pm 0.1}$ | $98.3 \pm 0.0$ | $98.3 \pm 0.0$ | $95.0 \pm 0.2$ | $\mathbf{97.2}$ |

Table 20: Benchmark: RotatedMNIST, Model selection: test-domain validation set (oracle).

| Algorithm | A | C | P | S | Avg |
|---|---|---|---|---|---|
| ERM | $83.7 \pm 0.5$ | $\mathbf{82.1 \pm 0.2}$ | $\mathbf{97.5 \pm 0.2}$ | $69.1 \pm 0.6$ | $\mathbf{83.1}$ |
| IRMv1 | $66.7 \pm 4.3$ | $68.5 \pm 1.6$ | $87.1 \pm 5.3$ | $67.7 \pm 2.0$ | 72.5 |
| IRMv1A | $83.5 \pm 0.2$ | $75.7 \pm 3.2$ | $96.4 \pm 0.3$ | $68.6 \pm 1.8$ | 81.0 |
| IRMv2 | $\mathbf{84.3 \pm 0.3}$ | $76.5 \pm 0.7$ | $96.8 \pm 0.1$ | $\mathbf{70.3 \pm 2.4}$ | 82.0 |

Table 21: Benchmark: PACS, Model selection: test-domain validation set (oracle).

| Algorithm | C | L | S | V | Avg |
|---|---|---|---|---|---|
| ERM | $\mathbf{98.4 \pm 0.1}$ | $65.1 \pm 1.4$ | $\mathbf{72.9 \pm 2.2}$ | $\mathbf{77.1 \pm 1.7}$ | $\mathbf{78.4}$ |
| IRM | $97.6 \pm 1.2$ | $61.9 \pm 0.6$ | $62.9 \pm 1.3$ | $73.0 \pm 0.3$ | 73.9 |
| IRMA | $98.0 \pm 0.1$ | $64.9 \pm 1.1$ | $71.8 \pm 0.8$ | $73.9 \pm 1.5$ | 77.1 |
| IRMv2 | $96.3 \pm 1.0$ | $\mathbf{67.1 \pm 0.1}$ | $70.9 \pm 1.3$ | $71.9 \pm 1.5$ | 76.5 |

Table 22: Benchmark: VLCS, Model selection: test-domain validation set (oracle).

| Algorithm | ColoredMNIST | RotatedMNIST | PACS | VLCS | Avg |
|---|---|---|---|---|---|
| ERM | $51.7 \pm 0.1$ | $96.7 \pm 0.0$ | $81.1 \pm 0.1$ | $\mathbf{78.8 \pm 0.4}$ | $\mathbf{77.0}$ |
| IRMv1 | $\mathbf{51.8 \pm 0.2}$ | $95.2 \pm 0.4$ | $78.6 \pm 1.0$ | $76.0 \pm 0.5$ | $75.4$ |
| IRMv1A | $50.9 \pm 0.1$ | $64.7 \pm 20.1$ | $80.9 \pm 0.0$ | $77.3 \pm 0.2$ | $68.4$ |
| IRMv2 | $50.8 \pm 0.4$ | $\mathbf{97.1 \pm 0.0}$ | $\mathbf{82.6 \pm 0.9}$ | $76.5 \pm 0.4$ | $76.8$ |

Table 13: Model selection: training-domain validation set.

| Algorithm | +90% | +80% | -90% | Avg |
|---|---|---|---|---|
| ERM | $30.4 \pm 13.4$ | $50.5 \pm 0.6$ | $9.9 \pm 0.0$ | $30.2$ |
| IRMv1 | $50.1 \pm 0.4$ | $\mathbf{60.6 \pm 7.3}$ | $\mathbf{30.0 \pm 14.1}$ | $\mathbf{46.9}$ |
| IRMv1A | $\mathbf{69.5 \pm 14.4}$ | $49.8 \pm 0.2$ | $10.0 \pm 0.1$ | $43.1$ |
| IRMv2 | $10.0 \pm 0.1$ | $36.4 \pm 3.0$ | $9.9 \pm 0.0$ | $18.8$ |

Table 14: Benchmark: ColoredMNIST, Model selection: leave-one-domain-out cross-validation.

| Algorithm | C | L | S | V | Avg |
|---|---|---|---|---|---|
| ERM | $97.5 \pm 0.8$ | $60.3 \pm 3.2$ | $70.1 \pm 4.2$ | $\mathbf{75.5 \pm 2.8}$ | $75.9$ |
| IRM | $93.3 \pm 2.7$ | $61.8 \pm 0.4$ | $72.9 \pm 0.9$ | $74.1 \pm 3.1$ | $75.5$ |
| IRMA | $79.2 \pm 12.8$ | $\mathbf{66.8 \pm 1.4}$ | $68.3 \pm 4.1$ | $73.5 \pm 1.3$ | $71.9$ |
| IRMv2 | $\mathbf{98.2 \pm 0.1}$ | $63.0 \pm 0.1$ | $\mathbf{74.4 \pm 1.2}$ | $69.9 \pm 0.2$ | $\mathbf{76.4}$ |

Table 17: Benchmark: VLCS, Model selection: leave-one-domain-out cross-validation.

| Algorithm | ColoredMNIST | RotatedMNIST | PACS | VLCS | Avg |
|---|---|---|---|---|---|
| ERM | $30.2 \pm 4.3$ | $96.0 \pm 0.4$ | $78.0 \pm 0.9$ | $75.9 \pm 0.7$ | $70.0$ |
| IRMv1 | $\mathbf{46.9 \pm 2.1}$ | $94.8 \pm 0.2$ | $72.9 \pm 3.0$ | $75.5 \pm 1.6$ | $\mathbf{72.5}$ |
| IRMv1A | $43.1 \pm 4.8$ | $62.4 \pm 21.6$ | $74.0 \pm 5.0$ | $71.9 \pm 2.8$ | $62.8$ |
| IRMv2 | $18.8 \pm 1.1$ | $\mathbf{97.0 \pm 0.1}$ | $\mathbf{82.5 \pm 1.0}$ | $\mathbf{76.4 \pm 0.3}$ | $68.7$ |

Table 18: Model selection: leave-one-domain-out cross-validation.

| Algorithm | +90% | +80% | -90% | Avg |
|---|---|---|---|---|
| ERM | $72.2 \pm 0.3$ | $72.6 \pm 0.4$ | $12.4 \pm 1.1$ | $52.4$ |
| IRMv1 | $\mathbf{72.5 \pm 0.3}$ | $\mathbf{72.9 \pm 0.1}$ | $\mathbf{63.3 \pm 6.6}$ | $\mathbf{69.6}$ |
| IRMv1A | $71.4 \pm 0.2$ | $72.8 \pm 0.3$ | $50.2 \pm 0.1$ | $64.8$ |
| IRMv2 | $70.6 \pm 1.2$ | $71.9 \pm 0.6$ | $20.9 \pm 0.9$ | $54.4$ |

Table 19: Benchmark: ColoredMNIST, Model selection: test-domain validation set (oracle).

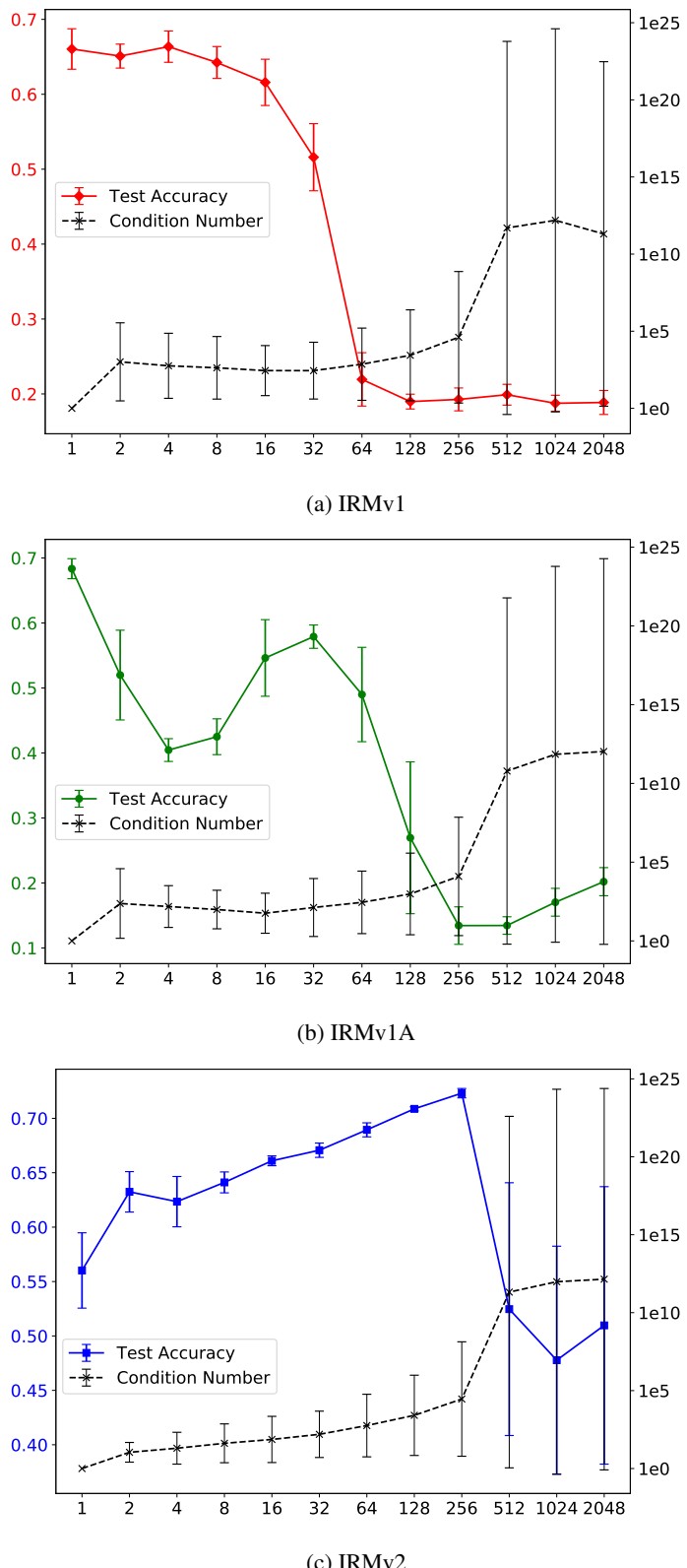

(a) IRMv1

(b) IRMv1A

(c) IRMv2

Figure 2: The figure depicts the evolution of test accuracy and the condition number of $\mathcal{I}(\varphi)$ in relation to the dimension of the representation.

