# OpenReview forum: "On Invariance Penalties for Risk Minimization"
_ICLR.cc/2022/Conference — ICLR 2022 Submitted_

### Official Review · Reviewer_AZJB · 2021-11-01

**Correctness:** 4
**Technical Novelty And Significance:** 2
**Empirical Novelty And Significance:** 2
**Recommendation:** 3
**Confidence:** 5

**Main Review:**

a)	Strength:
i.	This paper analyzes the counter-example of IRM and proposes a new invariance penalty to solve the OOD Generalization problem.
ii.	Practical insights on avoiding the failure case are provided for a better understanding of the counter-examples.
b)	Weaknesses:
i.	The proposed method is restricted to mean squared loss and cannot be extended to more general cases.
ii.	Although the role of I_e is demonstrated in Section 4.3, why the newly-proposed penalty is necessary and good is still unclear. And the derivation of the invariance penalty seems ad-hoc.
iii.	There lack of enough baselines in experiments. The authors only compare with ERM and IRM, while there are many other invariant learning methods, such as Rex[1], EIIL[2], HRM[3], etc.
iv.	Experimental results are not good enough to validate the effectiveness of the proposed method.

[1] D. Krueger, E. Caballero, J.-H. Jacobsen, A. Zhang, J. Binas, D. Zhang, R. Le Priol, and A. Courville, “Out-of-distribution generalization via risk extrapolation (rex),” ICML2021.
[2] E. Creager, J.-H. Jacobsen, and R. Zemel, “Environment inference for invariant learning,”ICML2021.
[3] J. Liu, Z. Hu, P. Cui, B. Li, and Z. Shen, “Heterogeneous risk minimization,”ICML 2021.


**Summary Of The Paper:**

This paper investigates the failure case of Invariant Risk Minimization and proposes a new invariance penalty for Out-of-Distribution Generalization. The authors provide some insights on how to avoid the potential failure of IRM based on the failure cases. And empirical results also validate the effectiveness of the proposed method.

**Summary Of The Review:**

This paper analyzes the counter-example of IRM and proposes a new invariance penalty based on that. Although some intuitions are provided, it is unclear why the proposed invariance penalty is necessary. And baselines, as well as the empirical results, are not enough to support the method.

---

> ### Author Response · Authors · 2021-11-16
> **Response to Reviewer AZJB**
>
> We thank the reviewer for their comments and noticing the intuition we provided on the counter-examples of IRM.
>
> In the updated manuscript, we have presented a result (Lemma 4) relating our invariance penalty to the difference of loss under an arbitrary classifier and the optimal one for cross-entropy loss. We hope that this result better motivate the use of our proposed penalty coefficient.
>
> In the Colored MNIST experiments (Figures 1 and 2) while IRMv1 fails with test accuracy of less than $0.2$ before the condition number of $\mathcal{I}_e$ reaches $10^4$, IRMv2 does not exhibit such failure and its test accuracy remains above $0.48$ for all condition numbers even when it surpasses $10^{10}$. This observation demonstrate that IRMv2 has the potential to alleviate the failure of IRMv1 in settings where $\mathcal{I}_e$ is ill-conditioned.

---

### Official Review · Reviewer_256p · 2021-11-01

**Correctness:** 3
**Technical Novelty And Significance:** 1
**Empirical Novelty And Significance:** 2
**Recommendation:** 3
**Confidence:** 4

**Main Review:**

In an earlier version of this review, I wrote that I felt that the objective proposed by this work is very similar to an existing linear regression objective from 2018, known as Anchor Regression [1]. The present paper proposes to consider linear regression *only*, which of course makes for an easier problem, but it also means there is lots of existing work in this area (for which there is insufficient discussion in the related work).

However, after further thought, I've realized that the two are only equal in quite specific circumstances. On the other hand, I have come across an observation that I think demonstrates the extent to which this work does not analyze their objective closely enough.

Consider the objective proposed by the paper:
$$\sum_e R_e(w^T\varphi) + \lambda ||I_e(\varphi)^{1/2} (w-w_e^*)||^2.$$
The first term is simply ERM. The second term, given features $X = \varphi(x)$, can be written $(w-w_e^*)^TX^TX(w-w_e^*)$. Consider what happens when we take the gradient of this objective with respect to our predictor $w$. We arrive at $2X^TX(w-w_e^*)$. Observe that this is precisely the gradient of linear regression, if we are assuming that our target is $Xw_e^*$ instead of the given target $y$. In other words, this is just linear regression where the targets are provided by a proxy classifier, as in two-stage least-squares. This tells us two things:

1. The objective is simply a weighted sum of the risk when predicting the true target $y$ and the "instrumental variable" target $Xw_e^*$ (equivalently, $w_e^{*T}\varphi(x)$). I believe this is nowhere mentioned in the paper and is worth exploring, if only because there doesn't seem to be any real understanding of what is going on with this objective.
2. I think my original impression that this was similar to Anchor Regression now appears pretty close to accurate, since Anchor Regression can also be seen as a trading off between ERM and a two-stage regression with instrumental variables.

Both of these points indicate to me that there is indeed something interesting going on with this objective, but this paper does not adequately explore the implications; nor does it properly discuss the previous related work (this paper takes existing work on logistic regression and applies it to linear regression, but there is a great deal of existing work in the causal literature on linear regression which is not discussed).

Returning to the paper itself, I find the empirical and theoretical results quite weak and I think this submission needs work even if it was a totally new idea. If there were some interesting new technical results to provide justification for this method in a different setting (such as classification) or under a different/more general model, it would certainly be noteworthy. However, I do not believe this to be the case: the setup Lemmas are simple linear algebra, and the primary theorem is almost identical to the related theorem in Rosenfeld et al. (furthermore, it only shows that it has the *same* linear environment complexity as IRM, rather than improving on it as the paper claims). The only new thing here is learning the feature embedder $\varphi$, but this paper provides no additional justification for when/why this should work---that is, all of the arguments for the benefits of this approach are agnostic to the actual quality of $\varphi$.

I also don't buy the claim that this would work in the non-linear setting; the result of Rosenfeld et al. is not specific to the type of loss, just the non-linearity of $\varphi$. The point behind that proof was that given only observations from the training environment, one can construct regions which contain all training points and on which the featurizer is perfect, and then make the featurizer arbitrarily bad elsewhere. In the absolute worst case, imagine a $\varphi = \varphi^*$ on all training points, and $\varphi = 0$ elsewhere. It seems clear to me that whatever claim the authors are making here with regards to the OOD performance of their predictor is not correct or is stated unclearly, because this solution would behave just like $\varphi^*$ at train time and give arbitrary error at test time. Even when observing the full environmental distributions, the fact that as we move away from the distribution mean the measure decays exponentially but the loss only grows quadratically means the same result would hold. I encourage the authors to revisit this section and work to clarify what precise mathematical claim they are making, and what this actually means in English.

Finally, *what is it with bolding empirical results which do not meet the bar for statistically significant improvement?* In Table 2, the confidence intervals of the performances of **all compared methods** overlap. Further, why aren't the standard deviations of the individual test accuracies reported over the five runs? There is zero indication that this method outperforms the ones to which it is being compared.

[1] Anchor regression: heterogeneous data meets causality. Dominik Rothenhäusler, Nicolai Meinshausen, Peter Bühlmann, Jonas Peters.


-------------------- Update after rebuttal --------------------

After discussion with the authors, I believe there are interesting aspects of this algorithm that are promising and worth exploring further. However, I still don't think the proposed method is analyzed or understood well enough, nor does there appear to be any substantial empirical or theoretical improvement here over existing works. I'm leaving my recommendation as is.

**Summary Of The Paper:**

The paper proposes a small modification to the IRMv1 objective which leads to an improvement over the original. They propose potential theoretical explanations for the improvement.

**Summary Of The Review:**

The idea behind this paper, while nice, is a repackaging of IV regression, and does not adequately analyze the implications of the objective (nor do they properly contextualize it within the great deal of similar related work in other fields). They provide no new theoretical justification (in fact their main technical result, very similar to another work, simply shows that this algorithm performs on par with the existing IRMv1). Further, their explication of performance in the non-linear setting is incomplete.

---

> ### Author Response · Authors · 2021-11-16
> **Response to Reviewer 256p**
>
> We thank the reviewer for their comments. In what follows, we provide detailed responses to the main comments, which we hope can clarify our contributions.
>
> **Rosenfeld et al.'s Setting:**
>
> As the reviewer points out, $\varphi_\epsilon$ is equal to the invariant representation $\varphi_{inv}$ on all the training points (assuming $\epsilon$ is appropriately chosen) and is different elsewhere. Rosenfeld et al. justify the relevance of this representation to IRM by showing that the invariance penalty of Arjovsky et al.'s can be made arbitrarily small. Using this together with the fact that the difference of risk under $\varphi_\epsilon$ and $\varphi_{inv}$ is small, they show that
>
>
> *IRMv1's objective (risk $+\ \lambda \ \cdot$ invariance penalty) for $\varphi_\epsilon$ is close to the optimal value of IRMv1 attained by the invariant representation.*
>
>
> For this observation to be true, one needs to assume that the penalty coefficient $\lambda$ is a constant. We show in Appendix C, that $\lambda_{\min} (E[\varphi_\epsilon(X)\varphi_\epsilon(X)^\top])\leq O(p_\epsilon)$. As Rosenfeld et al. show, the invariance penalty of IRMv1 is $O(p_\epsilon^2)$. Hence, by allowing $\lambda$ to be adaptive, e.g., $\lambda(\varphi) = 1 / \lambda_{\min}(E[\varphi_\epsilon \varphi_\epsilon^\top])^2$, we have that
>
>
> *The adaptive IRMv1’s objective (risk$+\ \lambda(\varphi)\ \cdot$  invariance penalty with adaptive penalty weight) for $\varphi_\epsilon$ may NOT be close to the optimal value attained by the invariant representation.*
>
>
> The Rosenfeld et al.'s setting is interesting and their formal analysis has provided valuable insights on the strengths and shortcomings of the practical implementation of IRM proposed by Arjovsky et al., which we drew inspiration from. However, the above observation brings into question the relevance of their counter example to the IRM principle (i.e., the possibility that  $\varphi_\epsilon$ is attained as a solution of IRM) beyond the IRMv1 relaxation with a fixed penalty coefficient.
>
> **Anchor Regression and Instrumental Variable:**
> We appreciate the reviewer bringing up anchor regression [1], which proposes a modification of the least squares estimation based on projection on the span of an "anchor", i.e., minimizing $||(I-\Pi_A)(Y-Xw)||^2+ \gamma|| \Pi_A (Y-Xw)||^2$ with $\Pi_A = A(A^\top A)^{-1} A^\top$ where $A$ is the matrix of observation of the anchor variable that is assumed to be an exogenous variable. Naturally, this assumption does not hold in our setting (i.e., for $\varphi(X)$).
>
> Thank you for observing the similarity of the gradient of IRMv2's invariance penalty and that of ERM. As shown in the paper (Lemma 2), the optimal $w$ for IRMv2 and ERM matches. However, as Reviewer PkSM points out the difference is on how $\varphi$ is being optimized.
>
> **Empirical Performance:**
> As demonstrated in Table 3 in the supplemental material, IRMv2 and IRMv1A (IRMv1 with adaptive penalty coefficient that is derived based on IRMv2) outperform all other methods (including ERM, IGA, ANDMask, and IRMv1) in 13 out of 18 experiments.

---

> > ### Comment · Reviewer_256p · 2021-11-18
> > **Followup on response**
> >
> > Thanks for your remarks. I think there are several points here that perhaps I didn't state clearly enough in my initial review, as your response doesn't really address some of my concerns.
> >
> > 1. I sort of understand now your point about the adaptive penalty, though this was not at all clear to me from my initial readthrough. I think this needs quite a bit of clarification still. However, I don't think this solves the problem, for two reasons.
> >
> > First of all, you've demonstrated that the specific counterexample given in Rosenfeld et al. is addressed here, but part of this may be due to the specifics of the construction, e.g. the fact that the featurizer sharply transitions from "full features" to "only invariant features". Secondly, this is specific to full observations of the environments, and it is straightforward to see that for a finite (sub-exponential in the dimension) number of samples, this would not occur with overwhelmingly high probability. Further, as pointed out by reviewer PkSM, the experimental results do not seem to clearly support your claim here.
> >
> > Here is where I think I was unable to follow your precise claim. In section 4.3.1, you wrote: "In other words, although the invariance penalty is arbitrarily small, the risk is also arbitrarily close to the risk of the invariant predictor." Does this not hold for the *training environments only*? Are you claiming that on a new *test environment*, the risk of the learned predictor will be the same as the invariant predictor? It sounds like you are claiming this but I don't see how this follows.
> >
> > These points are not 100% clear to me one way or the other, but I would be cautious when pointing out that one specific counterexample does not apply to not insinuate that this means the adaptive penalty "solves the problem". **This work still provides no positive guarantees for the proposed objective**, aside from Theorem 1 which, again, is exactly the same positive result as for IRM and other recent proposals.
> >
> > 2. Next, regarding Anchor Regression, I think perhaps you have not carefully considered the similarities here. If one defines the anchor variable as an indicator for the environment, then you would recover the present objective, where $w_e^*$ has only a single parameter. I think it's possible that by defining the anchor variables appropriately, you could allow for any number of parameters in this vector.
> >
> > But this is somewhat besides the point---I'm not focused on whether or not the objective here is *brand new* (it isn't), but on whether or not this paper properly contextualizes the objective within related work for linear regression. There is a great deal of newer work on invariance penalties for logistic regression in deep learning, but this work does not focus on logistic regression, nor does it focus on how to learn $\varphi$ (the deep featurizer). This work instead focuses on linear regression, and is agnostic to the quality of $\varphi$, which means it is very similar to existing work in the causality literature which does exactly the same thing. So I think there is quite a lot of missing discussion and comparison.
> >
> > 3. My point about the tradeoff was not saying that the gradient is the same as ERM. I was emphasizing that this objective is doing a weighted risk minimization between ERM and IV regression, similar to several existing works (e.g. Anchor Regression, see also [1]). The fact that this work does even mention this connection is, to me, an indicator of the degree to which the proposed objective is not understood.
> >
> > 4. The mean score in Table 3 is better, but it is not statistically significant. The HealthyGut results display exactly the same thing; the superiority of this method is within the margin of error.
> >
> > [1] A causal framework for distribution generalization. Rune Christiansen, Niklas Pfister, Martin Emil Jakobsen, Nicola Gnecco, Jonas Peters.

---

> > > ### Author Response · Authors · 2021-11-20
> > > **Response to Follow-up Comments**
> > >
> > > We thank the reviewer for their follow-up comments and for giving us the opportunity to clarify some of the resulting misconceptions and concerns. This will be very useful to future readers. We will address the concerns point-by-point.
> > >
> > > **From point \#1:** "I sort of understand now your point about the adaptive penalty, though this was not at all clear to me from my initial readthrough. I think this needs quite a bit of clarification still. However, I don't think this solves the problem, for two reasons."
> > >
> > > **Response:** We are glad our reply helped. In reply to your two remarks on the specifics of the structure of the Rosenfeld et al's counter-example, we agree that the structure is somewhat contrived. However, the specifics are not important to the central argument on the importance of the eigenstructure of $E[\varphi(X)\varphi(X)^\top]$. We edited the manuscript to better clarify this point and focus on the central argument.
> > >
> > > Briefly, *our message is $d_\varphi=d_x$ is a poor choice as one may expect that the invariant features are only a subset of the observed features* and, more precisely, the $\mbox{span}(\varphi_{inv}(X))$ is a strict subset of $\mbox{span}(X)$. Hence, if $\varphi:R^{d_x} \rightarrow R^{d_\varphi}$ with $d_\varphi>d_{inv}$, then $E[\varphi_{inv}(X)\varphi_{inv}(X)^\top]$ will be ill-conditioned. This is precisely the case for both of the said examples as for their respective proposed representations it holds that $d_\varphi = d_x$ and $x$ include spurious and environment dependent variables.
> > >
> > > Specifically, in response to your question: "In section 4.3.1, you wrote: "In other words, although the invariance penalty is arbitrarily small, the risk is also arbitrarily close to the risk of the invariant predictor." Does this not hold for the training environments only?'' We removed this specific line from the manuscript as it was initially meant to enhance clarity and not engender confusion.
> > >
> > > In response to "This work still provides no positive guarantees for the proposed objective, aside from Theorem 1 which, again, is exactly the same positive result as for IRM and other recent proposals", we note that all theoretical results on IRM are limited to linear settings with the exception of the counter-example of Rosenfeld et al., which we have discussed its scope and clarified its limitations in details both in the paper and in our responses.
> > >
> > > **From point \#2:** "I'm not focused on whether or not the objective here is brand new (it isn't), but on whether or not this paper properly contextualizes the objective within related work for linear regression."
> > >
> > > **Response:** We need to clarify to the reviewer that the work is not about linear regression. All experiments except Example 1 of LinearUnitTests are *classification tasks* including ColoredMNIST, RotatedMNIST, VLCS, PACS, HealthyGutTests, LinearUnitTests (Examples 2 and 3) in which the respective featurizers are parameterized by neural networks (e.g., ResNet-50). Similar to Arjovsky et al., we simply motivate our implementation of IRM through the linear case (as mentioned in reply to one of the above points).
> > >
> > > **From point \#3:** "I was emphasizing that this objective is doing a weighted risk minimization between ERM and IV regression, similar to several existing works (e.g. Anchor Regression, see also [1])."
> > >
> > > **Response:** We have added the citation to Anchor Regression for the interested reader.
> > >
> > > **From point \#4:** "The mean score in Table 3 is better, but it is not statistically significant. The HealthyGut results display exactly the same thing; the superiority of this method is within the margin of error."
> > >
> > > **Response:** The fact that some of the confidence intervals have a slight overlap does not imply that the results are not statistically significant, see for example [1]. To clarify this, we provide to you in the supplemental material the actual statistical significance, calculated as a $p$-value, which is the typical way in which statistical significance is defined. In particular, the improvement of IRMv1A over IRMv1 is statistically significant for 11 out of 18 experiments in LinearUnitTests experiments.
> > >
> > > Furthermore, an invariant model must exhibit similar performance on the test and training sets. By comparing Tables 3 and 5, one can observe that the training and test errors of IRMv1A and IRMv2 are far closer than that of IRMv1. In particular, IRMv1 completely overfits to the training set on Example 2 and 3 with (training, test) errors of $(0.03, 0.45)$ on Example 2 and $(0.00, 0.37)$ on Example 3. IRMv1A (training, test) error is $(0.08,0.28)$ on Example 2 and $(0.49, 0.30)$ on Example 3. IRMv2 (training, test) error is $(0.24,0.36)$ on Example 2 and $(0.30,0.40)$ on Example 3.
> > >
> > > [1] Austin, Peter C., and Janet E. Hux. "A brief note on overlapping confidence intervals." Journal of vascular surgery 36.1 (2002): 194-195.

---

> > > > ### Author Response · Authors · 2021-11-20
> > > > **Further Discussion on the Numerical Experiments**
> > > >
> > > > - **ColoredMNIST Experiment:** These experiments demonstrate the connection between the eigenstructure of $E[\varphi(X)\varphi(X)^\top]$ and invariance. As Figures 1 and 2 demonstrate IRMv1's test accuracy drops to 0.2 as $d_\varphi$ increases. On the other hand, the performance of IRMv2 is stable and even slightly increases up to $d_\varphi=256>d_x=196$. As we mentioned in our response to reviewer PkSM, the sharp drop of the test accuracy of IRMv1 coincides with an order of magnitude increase in the condition number of $E[\varphi(X)\varphi(X)^\top]$. This experiment, which clearly is statistically significant demonstrate (a) the practical relevance of the connection between the eigenstructure of $E[\varphi(X)\varphi(X)^\top]$ and the ability of the featurizer to recover an invariant representation (b) the $E[\varphi(X)\varphi(X)^\top]^{1/2}$ difference between the penalty of IRMv1 and IRMv2 can improve stability of IRM, which is not surprising in light of Lemmas 1 and 3.
> > > >
> > > > - **DomainBed Experiments:** *The improvement of IRMv2 over ERM and IRMv1 is statistically significant for RotatedMNIST and PACS experiments.* We note that as demonstrated in https://github.com/facebookresearch/DomainBed, all state-of-the-art domain generalization methods perform similarly on these classification tasks, which we mention in Section G.1.

---

> > > > ### Comment · Reviewer_256p · 2021-11-21
> > > > **Summary of what I've taken from this discussion**
> > > >
> > > > Ok, so now that we're on the same page, let me summarize the concerns I still have with this submission, and the reason I am maintaining my original recommendation. I want to emphasize again that I think this is a direction with real potential, but I just feel the current submission doesn't sufficiently study, justify, or explain any aspects of the proposed algorithm.
> > > >
> > > > * I'm glad to see that this note about the eigenstructure has been clarified. However, it is still being used as a vague justification for your proposed algorithm. I'm sure there are interesting things happening here, and perhaps the variable penalty term could results in improvements---but I think there is nothing substantial in this paper, and this discussion needs to be fleshed out significantly if the intent is for this to be a main focus of the paper (it is in fact alluded to in the title, no?)
> > > >
> > > > * If this is not the (or a) main focus of the paper, then the main focus must be the algorithm IRMv2. But again, there is little justification for this algorithm. "Eigenstructure" arguments aside, the only positive result for IRMv2 is the exact same theoretical justification for IRMv1 as provided by Rosenfeld et al. There is no technical contribution here, nor is the environment complexity actually *improving* over IRMv1. Perhaps this method does slightly better on linear unit tests, but it is not a substantial improvement, the significance is questionable, and it's on toy examples. I note that this is *even with the variable penalty*. Basically, if this method is better, there should be experiments demonstrating better performance on a variety of tasks. And if it's not so much better in practice, you should at least be able to give a theoretical demonstration of its superiority. This paper includes neither.
> > > >
> > > > One final point: I don't think you can run experiments on the entire DomainBed benchmark and then only point to statistically significant improvements on specific datasets. Did you do any sort of correction here for multiple testing? It doesn't seem like you did. I would recommend including some sort of correction in future submissions, if your intent is to rigorously argue for the superiority of this algorithm.

---

### Official Review · Reviewer_2SD7 · 2021-11-02

**Correctness:** 4
**Technical Novelty And Significance:** 2
**Empirical Novelty And Significance:** 2
**Recommendation:** 5
**Confidence:** 3

**Main Review:**

This paper is well written but the innovation is insufficient. The form of proposed invariant penalty is similar to the one of invariant penalty proposed in [1] and limited to the least-square loss. Similar theoretical result was obtained in [2] when  in the linear case. Moreover, considering that there have been many alternatives of IRM in recent literature, a detailed explanation should be given to fully illustrate the strength and weakness of the proposed method compared to the existing alternatives.

**Summary Of The Paper:**

This paper studies the problem of generalizing out of distribution (OOD), i.e., when data is not iid. Motivated by the error decomposition of the least-squares scheme, this paper proposes a novel invariance penalty that is directly related to risk. Based on the newly proposed penalty, this paper introduces another implementation of the IRM, named IRMv2. By characterizing the difference between the proposed penalty and the one in [1], this paper provides an adaptive rule for the choice of penalty parameter in the original IRMv1. Theoretical results guarantee that the proposed scheme can capture invariance when  in the linear case. The conditional number of Gram matrix of the data representation is analyzed.

**Summary Of The Review:**

Specific Comments
1）The similar theoretical result in this paper has been obtained in [2]. In other words, under the same condition, the theoretical result also holds for IRM. It confuses me whether the proposed method is essentially different from IRM.
2）It seems to me that the proposed invariance penalty doesn’t solve the problem of IRM. This paper doesn’t guarantee that the proposed IRMv2 can recover invariant predictor at least in the theoretical scenario where IRM fails.
3）I suggest that a detailed comparison with other alternatives for IRM should be given to better illustrate the strength and weakness of the newly proposed invariant penalty.
4）There are a few places that are not very rigorous. (1) In 5-th line of Algorithm 1,  is computed according to Eq. equation 4. However, the distribution is usually unknown and expectation can’t be computed. (2) In 6-th line of Algorithm1,  is computed according to Eq. equation 12 instead of Eq. equation 11.

[1] Martin Arjovsky, Léon Bottou, Ishaan Gulrajani, and David Lopez-Paz. Invariant risk minimization. arXiv preprint arXiv:1907.02893, 2019.
[2] Elan Rosenfeld, Pradeep Kumar Ravikumar, and Andrej Risteski. The risks of invariant risk minimization. In International Conference on Learning Representations, 2021. URL https://openreview.net/forum?id=BbNIbVPJ-42.

---

> ### Author Response · Authors · 2021-11-16
> **Response to Reviewer 2SD7**
>
> We thank the reviewer for their comments and we are glad that they find the paper well-written.
>
> 1 & 2. **Relationship to IRM:** IRM in its original form amounts to a constrained bi-leveled optimization problem, which is not practically feasible. In this paper, we revisit the failure mode of IRMv1 (the relaxation of IRM proposed by Arjovsky et al.) and relate the failure of the prominent counter examples in the literature to the Gram matrix of the data representation being ill-conditioned. Based on this insight, we propose an alternative implementation of the IRM principle as well as a modification of IRMv1 in the form of an adaptive penalty that has the potential to improve its performance as demonstrated in the LinearUnitTest experiments. We also empirically demonstrate in Figure 1 and 2 that IRMv2 outperforms IRMv1 by a significant margin when the condition number of the said Gram matrix exceeds $10^4$ in the Colored MNIST experiments.
>
> 3. **Comparison with other methods:** The main focus of this paper is to investigate the potential reasons for failure of IRMv1 and propose methods to alleviate such behavior. Hence, we limit the scope of the empirical evaluation to ERM and IRMv1.
>
> 4. **Unknown distributions:** We have provided details of the empirical estimation of the invariance penalty of IRMv2 in Appendix B of the updated paper.
>
> We also thank the reviewer for pointing out a typo in the algorithm pseudo-code. We have corrected it in the updated paper.

---

### Official Review · Reviewer_PkSM · 2021-11-05

**Correctness:** 3
**Technical Novelty And Significance:** 3
**Empirical Novelty And Significance:** 3
**Recommendation:** 6
**Confidence:** 4

**Details Of Ethics Concerns:**

NA.

**Main Review:**

## Major points

1. IRM penalties are most often studied alongside representations learnt by deep learning models, which are most often trained using some variant of stochastic gradient descent. Concretely, Arjovsky et al. 2019 proposed an unbiased stochastic estimator of the gradient of the original IRM penalty. However, to the best of my knowledge, the proposed approach as presented would in principle require computing the gradient of the ERM regressor $w^{*}(\Phi) = (\Phi^{T}\Phi)^{-1}\Phi^{T}y$ with respect to $\Phi$. Similarly, the adaptive rule to adjust the penalty weight for IRMv1 presented in Section 3.1. would in principle require estimating the smallest eigenvalue of $\Phi_{e}^{T}\Phi_{e}$, which might be difficult in a stochastic setting.

    I believe it would be ideal if the authors could address this limitations either theoretically (e.g. by proposing unbiased gradient estimators) or empirically (e.g. by proposing heuristic, possibly-biased gradient estimators and demonstrating good empirical performance nevertheless). At the very least, the issue of how to estimate the gradient of $L_{t}(\phi_{\theta_{t}})$ in Line 7 of Algorithm 1 in practice should be described in more detail. If it turns out that the proposed approach can only be used with (non-stochastic) gradient descent, this should at least be explicitly stated in the manuscript to ease the way for future work.

2. The proposed approach can only be used alongside the MSE loss. While this is acknowledged by the authors, which is appreciated, I do believe this to be an important practical limitation, as this loss is rarely used for supervised classification. Moreover, this makes the soundness of the adaptive rule in Section 3.1. questionable whenever IRMv1 is used in conjunction with e.g. the logistic loss.

3. The adaptive rule in Section 3.1. could be better justified. While the bound in Lemma 3 is to the best of my knowledge sound for the MSE loss, it is not explicitly stated how that motivates Eq. (15) exactly and what the theoretical implications are, if any. This is in my opinion important as it could be argued this adaptive rule is the contribution that has the most practical impact in performance according to the experiments in Section 5.

4. The authors make a point that there might be a link between ill-conditioning of the Gram matrix $\Phi_{e}^{T}\Phi_{e}$ and failure modes of IRM penalties, providing theoretically that some of the most well-known counterexamples to these penalties do exhibit ill-conditioning. However, the empirical results in Figure 2 in the appendix are somewhat conflicting. In particular, it would appear that the "phase transition" in which test accuracy sharply drops precedes the sharp increase in ill-conditioning particularly for IRMv1 and IRMv1A.

5. All in all, the experimental results do not convey a clear benefit of using IRMv2 vs IRMv1. The HealthyGutTests results in Table 2 are, as far as I can tell, well within the margin of error. Most importantly, it would appear IRMv1 (and the IRMv1A variant) dominate IRMv2 in the LinearUnitTests experiments.

## Typos

Proofs in the appendix often refer to the $\mathcal{l} - 2$ ball instead of $\mathcal{l}_{2}$.

**Summary Of The Paper:**

In this paper, the authors propose an alternative Invariant Risk Minimization (IRM) penalty for the particular case of a linear output layer trained with a mean-squared error loss. In this particular case, the original formulation can be understood as penalizing the squared Euclidean norm of the output layer's gradient, that is,

$$\rho_{e}^{\mathrm{IRMv1}}(\Phi, w) := \vert\vert \Phi_{e}^{T}(\Phi_{e} w - y_{e}) \vert\vert_{2}^{2} = \vert\vert \Phi_{e}^{T}\Phi_{e}(w -w_{e}^{*})\vert\vert_{2}^{2},$$

where $\Phi_{e} \in \mathbb{R}^{n_{e} \times d}$  is a matrix with the (learnt) data representation for environment $e$, $y_{e}$ the corresponding targets and $w_{e}^{*} = (\Phi_{e}^{T}\Phi_{e})^{-1}\Phi_{e}^{T}y_{e}$ the optimal, unregularized linear regressor. Witth this in mind, this paper proposes to use instead:

$$\rho_{e}^{\mathrm{IRMv2}}(\Phi, w) := \vert\vert (\Phi_{e}^{T}\Phi_{e})^{1/2}(w -w_{e}^{*})\vert\vert_{2}^{2},$$

which interestingly can be trivially shown to lead to an optimal classifier $w^{*}$ identical to that of the ERM formulation. Crucially, however, the authors propose to explicitly backpropagate through the dependence of this classifier on the data representations $\Phi_{e}$ when minimizing a cost function that incorporates $\rho_{e}^{\mathrm{IRMv2}}(\Phi, w)$.

The authors prove that, under similar non-degeneracy conditions as those assumed when analyzing other IRM penalties, $\rho_{e}^{\mathrm{IRMv2}}(\Phi, w)$ will either find an invariant data representation (or a trivial classifier) for a linear SEM as in Rosenfeld et al. 2021. Furthermore, the authors argue that there might be a connection between ill-conditioning of the Gram matrix $\Phi_{e}^{T}\Phi_{e}$ and failure modes of IRM penalties and propose a heuristic rule to adapt the penalty weight in the original IRM formulation.

**Summary Of The Review:**

Out-of-distribution generalization is one of the most pressing, largely open problems in the field. Concretely, the IRM framework has recently emerged as a promising line of research in this direction, but its characterization is far from conclusive both from a theoretical and a practical standpoint. Thus, I consider the paper's topic timely and of great relevance to the field.

I believe the manuscript  explores an interesting idea and is technically sound, albeit not without what I consider to be important limitations, namely,
+ The proposed approach is only applicable to the MSE loss.
+ It is unclear whether it would be sound to use the proposed penalty in a stochastic gradient-based optimization setting, as it requires backpropagating through the optimal linear regressor, which depends on all data.
+ The experimental results do not provide compelling evidence that the proposed penalty is statistically significantly superior to the original formulation.

All in all, I believe IRM to be a promising yet not fully mature idea with many crucial open questions remaining to be answered. Thus, I consider exploratory ideas such as those presented in this article to be worthwhile despite potential shortcomings that might need to be revisited by future work. Hence, despite the aforementioned limitations, I lean towards supporting acceptance of the manuscript.

---

> ### Author Response · Authors · 2021-11-16
> **Response to Reviewer PkSM**
>
> We appreciate the reviewer's positive feedback and their thorough review of our paper. The reviewer raises important questions, which we discuss below.
>
> 1.  **Estimation and Bias:** We have provided the unbiased estimators of IRMv2's invariance penalty using data from different mini-batches in Appendix B of the updated paper. Regarding the estimation of the minimum eigenvalue of $\Phi_e^\top \Phi_e$, first note that its introduction in the adaptive penalty coefficient (IRMv1A) is to balance the trade-off between the risk and invariance penalty, especially in the regime in which $\Phi_e^\top \Phi_e$ transitions into being ill-conditioned. As it is proposed based on upper bounds (Lemmas 3 and 4), we speculate that a coarse estimate is sufficient to capture that trend. Second, we note that the computation burden of the minimum eigenvalue would not be significant (in comparison to the training of large models, e.g., ResNet-50 in DomainBed experiments). More specifically, $\Phi_e^\top \Phi_e$ is a $d_\varphi\times d_\varphi$ matrix where $d_\varphi$ is a user chosen variable and often $d_\varphi\ll d_x$. In particular, in implementation of IRMv1 for classification, $d_\varphi = K-1$ where $K$ is the number of classes.
>
> 2 & 3. **Relation to other losses:** In the updated paper, for cross-entropy loss, we have provided a result (Lemma 4) that upper bounds the risk difference of two classifiers in terms of our penalty. This upper bound helps justifying the use of the adaptive penalty in conjunction with cross-entropy loss.
>
> 4. **Ill-conditioned Gram matrix:** We thank the reviewer for taking time to read the supplemental material. The $y$ axis is exponential. So in the sharp drop of test accuracy of IRMv1 from $d_\varphi =32$ to $d_\varphi = 128$, the condition number increases from less than 100 to over 2000.
>
> 5. **Empirical performance:** As the reviewer points out IRMv1A, which relies on our adaptive approach in choosing the penalty coefficient, dominates all methods in the majority of the LinearUnitTests experiments.

---

### Decision · Program_Chairs · 2022-01-20

**Decision:**

Reject

**Comment:**

This manuscript was the object of a rich and lengthy discussion. The AC also felt compelled to read the paper in details and discussed it further with the SAC.

The authors did a thorough job at addressing some of the reviewers points. The added results on cross-entropy loss and additional discussion, as well as the points made in "Further Discussion on the Numerical Experiments" are very much appreciated.

However, significant concerns remain on establishing connections with prior work, including related ideas on invariance from the causality literature, so as to gain deeper understanding of the implications of the proposed objective. We also strongly encourage the authors to further work on strengthening their theoretical analysis to clearly demonstrate the value of the proposed approach.

The proposed formulation is certainly thought provoking and we urge the authors to pursue their work in view of the above comments.